# Disordered regions and folded modules in CAF-1 promote histone deposition in *Schizosaccharomyces pombe*

Fouad Ouasti[1†], Maxime Audin[1†], Karine Fréon[2], Jean-Pierre Quivy[3], Mehdi Tachekort[1], Elizabeth Cesard[1], Aurélien Thureau[4], Virginie Ropars[1], Paloma Fernández Varela[1], Gwenaelle Moal[1], Ibrahim Soumana Adamou[2], Aleksandra Uryga[2], Pierre Legrand[4], Jessica Andreani[1], Raphaël Guerois[1], Geneviève Almouzni[3], Sarah Lambert[2]*, Francoise Ochsenbein[1]*

[1]Université Paris-Saclay, CEA, CNRS, Institute for Integrative Biology of the Cell (I2BC), Institute Joliot, Gif-sur-Yvette, France; [2]Institut Curie, PSL Research University, CNRS UMR 3348, INSERM U1278, Université Paris-Saclay, Equipe labellisée Ligue contre le Cancer, Orsay, France; [3]Institut Curie, PSL Research University, CNRS, Sorbonne Université,CNRS UMR3664, Nuclear Dynamics Unit, Équipe Labellisée Ligue contre le Cancer, Paris, France; [4]Synchrotron SOLEIL, HelioBio group, l'Orme des Merisiers, Saint-Aubin, France

*For correspondence: sarah.lambert@curie.fr (SL); francoise.ochsenbein@cea.fr (FO)

[†]These authors contributed equally to this work

Competing interest: The authors declare that no competing interests exist.

## eLife assessment

This **important** study advances our understanding of the machinery that couples DNA synthesis with the deposition of histone proteins onto newly synthesized DNA. A **convincing** array of experiments combines NMR, protein biochemistry, and in vivo analyses of Chromatin Assembly Factor-1 of fission yeast. The work is of interest to researchers in the field of chromosome/chromatin biology as well as epigenetics.

**Abstract** Genome and epigenome integrity in eukaryotes depends on the proper coupling of histone deposition with DNA synthesis. This process relies on the evolutionary conserved histone chaperone CAF-1 for which the links between structure and functions are still a puzzle. While studies of the *Saccharomyces cerevisiae* CAF-1 complex enabled to propose a model for the histone deposition mechanism, we still lack a framework to demonstrate its generality and in particular, how its interaction with the polymerase accessory factor PCNA is operating. Here, we reconstituted a complete *Sp*CAF-1 from fission yeast. We characterized its dynamic structure using NMR, SAXS and molecular modeling together with in vitro and in vivo functional studies on rationally designed interaction mutants. Importantly, we identify the unfolded nature of the acidic domain which folds up when binding to histones. We also show how the long KER helix mediates DNA binding and stimulates *Sp*CAF-1 association with PCNA. Our study highlights how the organization of CAF-1 comprising both disordered regions and folded modules enables the dynamics of multiple interactions to promote synthesis-coupled histone deposition essential for its DNA replication, heterochromatin maintenance, and genome stability functions.

## Introduction

In eukaryotes, genomic DNA is packaged in a dynamic nucleoprotein complex, the chromatin, which protects DNA and regulates its accessibility. The fundamental repeat unit of chromatin, the nucleosome core particle, comprises 146 bp of DNA wrapped around a histone octamer including a tetramer of histone H3−H4 flanked by two dimers of H2A−H2B (*Luger et al., 1997*). Histone chaperones are critical players in ensuring histone traffic and deposition. Without energy consumption, they escort histones, facilitate their transfer and deposition on DNA, and provide links with DNA based-processes such as DNA replication, repair, and gene transcription (*Gurard-Levin et al., 2014*). In line with these key properties, perturbations of histone chaperones are associated with defects in genome and epigenome maintenance and function as found in cancer, aging, and viral infections (*Yi and Kim, 2020*; *Sultana et al., 2021*; *Ray-Gallet and Almouzni, 2022*). Discovered over 30 years ago (*Smith and Stillman, 1989*), conserved in all eukaryotes (*Loyola and Almouzni, 2004*), the histone chaperone <u>C</u>hromatin <u>A</u>ssembly <u>F</u>actor 1 (CAF-1) is central and unique in promoting the deposition of replicative histones H3−H4 in a manner coupled to DNA synthesis, that is during DNA replication and repair, and is also involved in heterochromatin maintenance (see *Ridgway and Almouzni, 2000*; *Gurard-Levin et al., 2014* for review). The unique feature of CAF-1 is that it provides a link with DNA synthesis via its association with the trimeric DNA polymerase processivity factor, proliferation cell nuclear antigen (PCNA), through <u>P</u>CNA <u>I</u>nteracting <u>P</u>rotein motifs (PIP; *Martini et al., 1998*; *Shibahara and Stillman, 1999*; *Moggs et al., 2000*; *Zhang et al., 2000*; *Rolef Ben-Shahar et al., 2009*; *Pietrobon et al., 2014*; *Gopinathan Nair et al., 2022*), but the precise function of this interaction remains to be examined.

CAF-1 comprises three subunits (*Figure 1A*; *Smith and Stillman, 1989*; *Kaufman et al., 1997*; *Dohke et al., 2008*). While progress in uncovering its molecular/genetic properties derives from work in *Saccharomyces cerevisiae,* and biochemical work in human cells, there is still a lack of atomic information for the complex. In *S. cerevisiae*, CAF-1 is a hetero-trimer that binds to a single dimer (*Mattiroli et al., 2017a*, *Sauer et al., 2017*). The current model states that histone binding induces a conformational rearrangement promoting its interaction with the DNA. Two complexes must co-associate to ensure the deposition of H3−H4 tetramers on DNA in the first step of nucleosome assembly (see *Sauer et al., 2018* for review). Two domains of the large subunit Cac1 contribute to DNA binding (*Zhang et al., 2016*; *Sauer et al., 2017*), the conserved low complexity region called KER (for Lys, Glu, and Arg rich) and the C-terminal <u>W</u>inged <u>H</u>elix <u>D</u>omain (WHD). These features are conserved in human CAF-1 (*Gopinathan Nair et al., 2022*). However, a comprehensive and dynamic view of the CAF-1 complex is still missing, and most critically understanding how the functional domains cooperate to ensure efficient histone deposition coupled to DNA synthesis remains to be determined. Moreover, the degree of conservation of these properties across species and possible connections with heterochromatin regulation elements such as histone H3K9 trimetylation and HP1 recruitment remain unknown.

To tackle these issues, we isolated the fission yeast complex (Pcf1-Pcf2-Pcf3) and investigated its binding mode with its three main partners, histones H3−H4, DNA, and PCNA. Based on novel structural insights, we designed targeted mutations to specifically identify the key features promoting Pcf1 interactions with DNA, PCNA and histones H3−H4. We next monitored functional impacts of these mutations in vitro and in vivo by analysing the phenotypes of the corresponding mutants in fission yeast. We show that disordered or partly disordered segments of the large subunit of CAF-1 are key for interactions with H3−H4 and DNA, underscoring the dynamic nature of the binding interface between CAF-1, histones H3−H4, and DNA. Upon histone binding, the acidic domain (ED) of the large subunit folds and this conformational changes impact CAF-1-PCNA interaction in vivo. We propose that such conformational changes upon histone binding contribute to PCNA/CAF1 recycling during DNA replication. We further show that PCNA binding accelerates nucleosome assembly in vitro and is also essential for the proper targeting of the complex to the chromatin in vivo. Finally, we found that the WHD C-terminal domain does not bind DNA but allows to uncouple the unique functions of CAF-1 in replication-dependent chromatin assembly, genome stability and heterochromatin maintenance. We suggest that this domain contributes to specify CAF-1 functions.

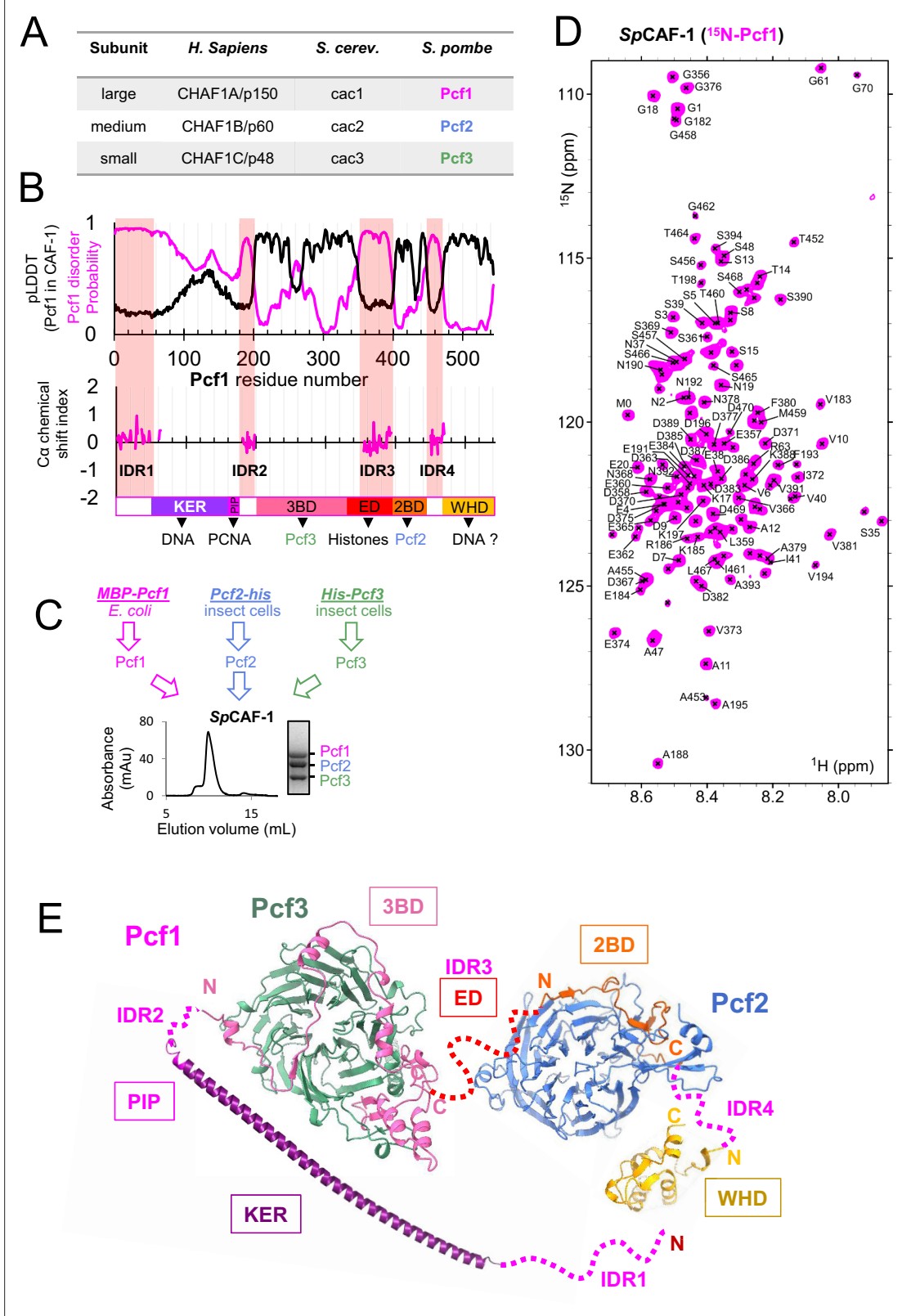

**Figure 1.** The large *SpCAF-1* subunit includes four Intrinsically Disordered Regions (IDR). (**A**) Names for the large, medium and small subunits of CAF-1 in *H. sapiens*, *S. cerevisiae*, and *S. pombe*. (**B**) Upper panel: The magenta line shows the predicted disorder of Pcf1 (spot disorder software) and the black line the Cα Local Distance Difference Test (pLDDT) calculated for Pcf1 residues by the AlphaFold2 model of the full *SpCAF-1* complex. Lower panel: Cα chemical shift index calculated for the 101 assigned residues. This Cα chemical shift index is consistent with their disordered nature. The four

**Figure 1 continued**

IDR regions are highlighted with pink semi-transparent vertical bars. The predicted domains of Pcf1 are labeled. (**C**) General strategy for the production of *Sp*CAF-1. The lower panel shows the purification SEC profile and the SDS-PAGE purity of the sample. (**D**) $^1$H-$^{15}$N SOFAST-HMQC spectrum of the FL *Sp*CAF-1 complex composed of uniformly labeled $^{15}$N-Pcf1 and unlabeled Pcf2 and Pcf3 (*Sp*CAF-1($^{15}$N-Pcf1)). The assigned signals are labeled. (**E**) AF2 model of the *Sp*CAF-1 complex. The four IDR segments are shown with a dashed line. The relative orientation of the four modules is arbitrary.

The online version of this article includes the following source data and figure supplement(s) for figure 1:

**Figure supplement 1.** Sequence and constructions of Pcf1, the large subunit of *Sp*CAF-1.

**Figure supplement 2.** Reconstitution and structural analysis of *Sp*CAF-1.

**Figure supplement 2—source data 1.** Uncropped SDS Page gels for the SEC analysis of recombinant Pcf1, Pcf2, and Pcf3 proteins purified separately presented in *Figure 1—figure supplement 2A, B* (lower panels).

**Figure supplement 3.** Structure prediction of the module Pcf1(403-450)-Pcf2(1-453) from *Sp*CAF-1.

**Figure supplement 4.** Structure prediction of the module Pcf1(200-335)-Pcf3(1-408) from *Sp*CAF-1.

**Figure supplement 5.** Experimental and fitted SAXS data for SpCAF-1 and *Sp*CAF-1−H3−H4.

## Results

### Global organization of the full-length *Sp*CAF-1 complex

The large subunit of CAF-1, present in all major groups of eukaryotes, exhibits significant sequence divergence (16% sequence identity between *Sp*Pcf1 and *Sc*Cac1 and 21% sequence identity between *Sp*Pcf1 and *Hs*CHAF1A/p150, *Figure 1—figure supplement 1A*). Given this high sequence divergence, conserved biochemical properties between *Sc*CAF-1 and *Sp*CAF-1 should reveal important functional features. Of particular interest, *Schizosaccharomyces pombe* heterochromatin shares with human a regulation based on the recruitment of Swi6, the heterochromatin function 1 (HP1) orthologue, via histone H3K9 trimethylation. From sequence alignments, the six main conserved regions previously proposed to contribute to the nucleosome assembly activity of CAF-1 can be inferred in *Sp*Pcf1 sequence, a KER domain, a single PIP motif, an acidic domain (ED domain), the domains predicted to bind Pcf2 (2BD) and Pcf3 (3BD) and a C-terminal WHD domain (*Figure 1—figure supplement 1B*). Although *Sp*Pcf1 is shorter than *Sc*Cac1 and *Hs*CHAF1A/p150 (544 residues instead of 606 and 956, respectively) its sequence includes a remarkable high abundance of predicted Intrinsically Disordered Regions (IDRs; *Figure 1B*). These IDRs include the predicted histone-binding domain (Pcf1 ED), the PCNA (PIP motif) and the DNA binding domain (Pcf1 KER).

We produced and purified the three subunits of *Sp*CAF-1 separately from bacteria and insect cells (*Figure 1C*, materials & methods). When isolated, both Pcf2 and Pcf3 are monomeric while Pcf1 forms large soluble oligomers. Mixing the subunits by pairs, we observed by size exclusion chromatography (SEC), stable complexes for Pcf1-Pcf2 and Pcf1-Pcf3 (*Figure 1—figure supplement 2A*). Pcf2 and Pcf3 did not interact with each other (*Figure 1—figure supplement 2B*) suggesting that the large subunit Pcf1 mediates the complex assembly. We next reconstituted and isolated the recombinant full-length (FL) *Sp*CAF-1 complex by SEC (*Figure 1C*). An experimental molecular weight of 179 kDa was calculated using small angle X-ray scattering (SAXS). Assuming an accuracy of around 10% with this method (*Rambo and Tainer, 2013*), this value is consistent with a 1:1:1 stoichiometry for the CAF-1 complex (calculated MW 167 kDa; *Figure 1—figure supplement 2C*). In addition, the position of the maximum for the dimensionless Kratky plot was slightly shifted to higher values in the y and x axis compared to the position of the expected maximum of the curve for a fully globular protein (*Figure 1—figure supplement 2D*). This shows that the complex was globular with a significant flexibility.

To determine the extent of disorder in the large subunit of *Sp*CAF-1, Pcf1 was produced with uniform $^{15}$N (or $^{15}$N-$^{13}$C) labeling. The CAF-1($^{15}$N-Pcf1) complex with unlabeled Pcf2 and Pcf3 was reconstituted and SEC-purified. Given the size of this complex (167 kDa), we expected that only amide signals from residues in long disordered regions could be observed by nuclear magnetic resonance (NMR) spectroscopy. The $^{15}$N-$^1$H spectrum shows about 140 amide signals, revealing that up to a quarter of Pcf1 residues are intrinsically disordered in the full *Sp*CAF-1 complex (*Figure 1D*). These residues are located in four continuous segments of Pcf1 and define Intrinsically Disordered Regions that we labeled IDR1 to IDR4 (*Figure 1B*). IDR1 corresponds to the ~50 N-terminal residues of the protein, IDR2 (181-198) is located between the PIP motif and the 3BD region, IDR3 (355-394) overlaps a large segment of the acidic ED domain and IDR4 (451-470) is located between the 2BD region and

the C-terminal WHD domain. The boundaries of the four IDRs are in agreement with the segments of Pcf1 predicted to harbor disorder with a high probability (*Figure 1B*).

We next built a model of the *Sp*CAF-1 complex using the AlphaFold2 multimer software (AF2) with one copy of each FL protein (*Figure 1E*, *Figure 1—figure supplements 2–4*). The model is consistent with our biochemical data showing that Pcf1 mediates the complex assembly. Also, in agreement with their disordered nature, low values around 0.2–0.3 of the local quality of the model as calculated by the Local Distance Difference Test (pLDDT) were obtained in the four IDR segments with a remarkable match for the delimitations of the four IDR segments by pLDDT values and NMR data (*Figure 1B*). Accordingly, these segments are symbolized with a dashed line in *Figure 1E*. In contrast, significantly high pLDDT was obtained for the 3BD, 2BD and WHD domains of Pcf1 and for the two subunits Pcf2 and Pcf3 (*Figure 1B*, *Figure 1—figure supplement 2E*). These data allowed to identify four independent modules, not predicted to interact with each other. The first module corresponds to the KER domain of Pcf1, predicted to form a long helix ending by the PIP motif. The second module contains the Pcf2 subunit, composed of 7 WD repeats arranged in a circular fold, and a segment of Pcf1 corresponding to the 2BD domain forming three short beta strands and a short helix (*Figure 1—figure supplement 3*). In the third module, the 3BD domain of Pcf1 composed of seven helices and three beta strands establishes a large interface with Pcf3, composed of seven circular WD repeats (*Figure 1—figure supplement 4*). The fourth domain is the WHD domain. We next used these models to fit our SAXS data allowing flexibility between the four modules. The best model fitted the experimental data with a high accuracy and is in agreement with a relatively globular complex. Superimposing the generated models did not define a unique orientation between the four modules, suggesting that the complex has an inter-module flexibility (*Supplementary file 1a*, *Figure 1—figure supplement 5A–B*).

Taken together, our findings indicate that the large subunit Pcf1 mediates the (1:1:1) complex assembly. Pcf1 includes four IDR, and can organize its key regions (KER, PIP, 3BD, ED, 2BD, and WHD) allowing them to be exposed and bound by Pcf2 and Pcf3 simultaneously.

## In the FL *Sp*CAF-1 complex, the acidic ED domain is disordered but folds upon histone binding

We next investigated the interaction of *Sp*CAF-1 with histones H3−H4. A stable complex was isolated by SEC at low (150 mM NaCl) and high (1 M NaCl) salt concentrations, confirming that the reconstituted *Sp*CAF-1 complex tightly binds histones (*Figure 2A*). SAXS measurements at low salt allowed to calculate an experimental molecular weight of 193 kDa for this complex, showing that *Sp*CAF-1 binds a dimer of histones H3−H4 (*Figure 1—figure supplement 2C*). In addition, these data are compatible with a more extended shape compared to *Sp*CAF-1 alone (*Figure 1—figure supplement 2D*).

Addition of histones to *Sp*CAF-1($^{15}$N-Pcf1) led to a drastic decrease in intensity of the NMR signal specifically for residues in the IDR3 segment (*Figure 2B*, *Figure 2—figure supplement 1A*). To further characterize this domain, we designed a short construct of Pcf1 (325-396), called Pcf1_ED, corresponding to the IDR3 segment extended in its N-terminus with the conserved acidic segment (325-355; *Figure 2C*, *Figure 1—figure supplement 1B*). We confirmed the fully disordered nature of Pcf1_ED by NMR (*Figure 2—figure supplement 1B–C*). Signals corresponding to residues 355–394 (IDR3) remarkably overlap in the spectra of Pcf1_ED and *Sp*CAF-1($^{15}$N-Pcf1; *Figure 2—figure supplement 1A*), showing that this segment was fully flexible in *Sp*CAF-1, and did not interact with other regions of the complex. Upon binding of unlabeled histones H3−H4, we observed the vanishing of almost all NMR signals of $^{15}$N Pcf1_ED as in the full *Sp*CAF-1 complex (*Figure 2D*). In contrast, only a shorter fragment (338-347) vanished upon addition of Asf1−H3−H4−Mcm2(69-138), a histone complex preformed with two other histone chaperones, Asf1 and Mcm2, known to compete with CAF-1 for histone binding (*Sauer et al., 2017*) and whose histone binding modes are well established (*Figure 2E*; *Huang et al., 2015*; *Richet et al., 2015*). This finding underscores a direct competition between residues (325-338) and (349-396) within the ED domain and Asf1/Mcm2 for histone binding. Fully consistent with this NMR competition experiment, the segment (353-385) of Pcf1_ED domain was predicted by AlphaFold2 to interact with histones H3−H4 through the same surface as the one bound by Mcm2 (*Figure 2F*, *Figure 2—figure supplement 2A*). Two highly conserved positions in Pcf1, L359 and F380, are thus proposed to mediate histone H3−H4 binding in the same region as Mcm2 (*Figure 2F*). We next used these AlphaFold2 models to fit the SAXS curve of the

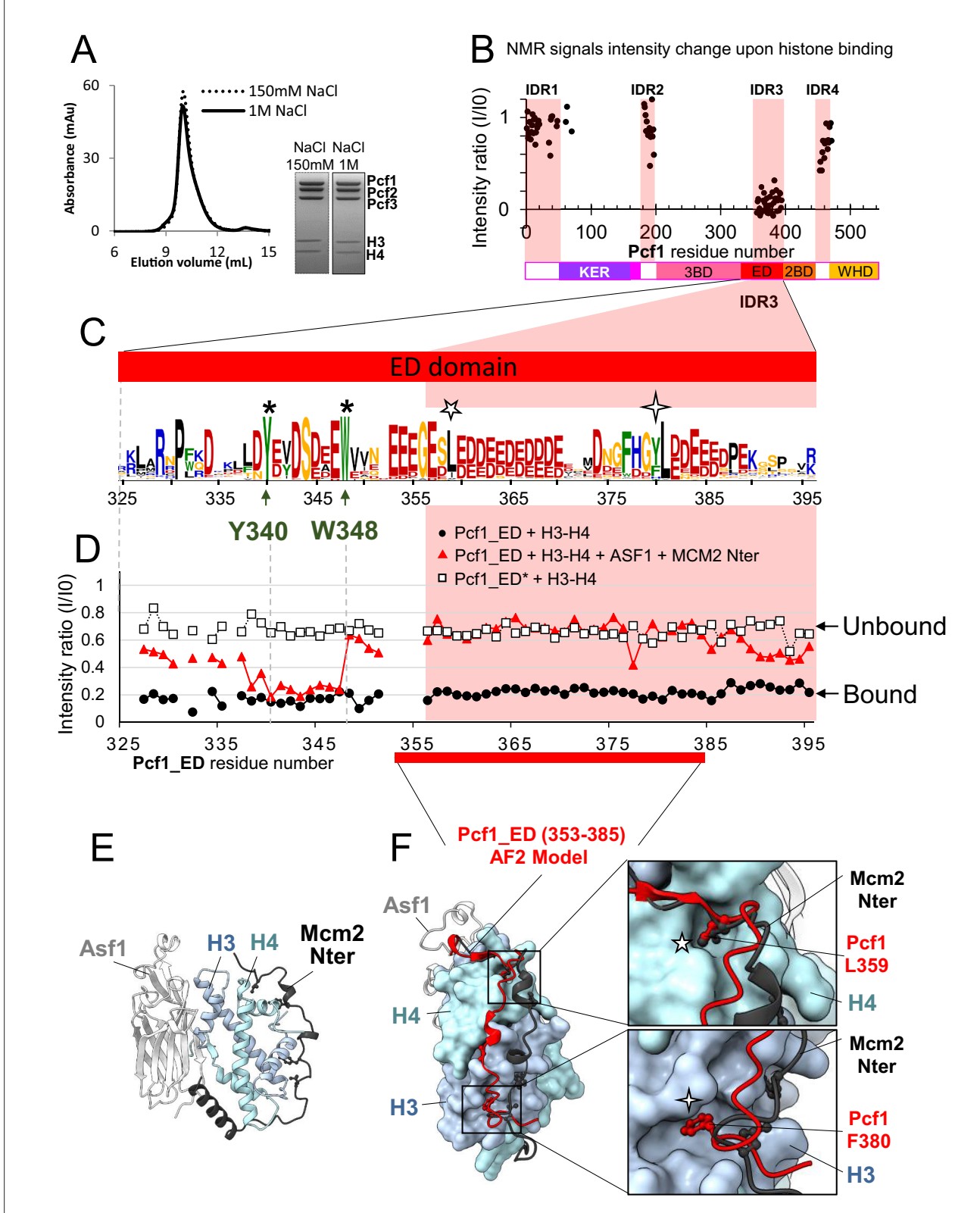

**Figure 2.** The acidic ED domain binds histones alone and in the full *Sp*CAF-1 complex. (**A**) SEC profile and the SDS-PAGE purity of *Sp*CAF-1−H3−H4 histones at 150 mM NaCl and 1 M NaCl. (**B**) Mapping of the interaction between *Sp*CAF-1($^{15}$N-Pcf1) and *Sp*H3−H4 histones, using the intensities ratio (I/I0), where I and I0 are the intensity of the signals $^{1}$H-$^{15}$N SOFAST-HMQC spectra before and after addition of histones, respectively. (**C**) Sequence Logo of the ED domain generated with a large data set of Pcf1 homologues. The position of the two conserved residues Y340 and W348, mutated in ED*

*Figure 2 continued on next page*

*Figure 2 continued*

are indicated with stars and conserved Pcf1 L359 and F380 residues with five and four branch stars respectively. (**D**) Mapping of the interaction between Pcf1_ED or Pcf1_ED* with *Sp*H3–H4 histones using the intensities ratio (I/I0) as in b. Histones were added alone or previously complexed with histones chaperones. (**E**) Cartoon representation of the complex between human histones H3–H4 (light blue and cyan), Asf1 (light grey) and Mcm2 (dark grey) (PDB: 5BNX). (**F**) AlphaFold2 model of Pcf1 (353-385) (as red cartoon), corresponding to the segment of the ED domain indicated in red, in complex with histones H3–H4 (light blue and cyan surface) superimposed with Mcm2 and Asf1 as in panel E. The two insets represent zoomed views of the sidechains of the conserved Pcf1 L359 and F380 residues (red sticks) binding into H4 and H3 pockets, respectively. The same four and five branch stars are used to label these positions in the logo panel C.

The online version of this article includes the following source data and figure supplement(s) for figure 2:

**Source data 1.** Uncropped SDS PAGE gels presented in *Figure 2A* and *Figure 2—figure supplement 2B*.

**Figure supplement 1.** The acidic ED domain binds histones alone and in the full CAF-1 complex.

**Figure supplement 2.** Structure prediction of the module Pcf1(352-383)–H3–H4 from *Sp*CAF-1 and purification of reconstituted CAF-1 complexes WT and mutants.

*Sp*CAF-1–H3–H4 complex allowing reorientation of the different modules (*Figure 1—figure supplement 5C–D*). Remarkably, all generated models show a significant exclusion of the KER domain from the complex, suggesting that the KER domain of *Sp*CAF-1 becomes more accessible upon histone binding.

The NMR competition experiment also reveals that an additional region of Pcf1_ED domain (338-351) is involved in the interaction with H3–H4 but is not competing with the Asf1-Mcm2 module (*Figure 2D*). In order to alter the interaction of the ED domain with histones without modifying its charge and without interfering with Asf1 or Mcm2 binding, we identified from sequence alignments in this segment (*Figure 2C*), two invariant hydrophobic residues, Y340 and W348, that were mutated into alanines (mutant called ED*, *Figure 1—figure supplement 1B*). As expected, the isolated Pcf1_ED* domain showed almost no histone binding as observed by the intensity of $^1$H-$^{15}$N NMR signals (*Figure 2D*). We next monitored the impact of the ED* mutations in the context of the full *Sp*CAF-1 complex. To do so, the mutations Y340A-W348A were introduced in the full length Pcf1, and the complex reconstituted with the uniformly $^{15}$N labeled Pcf1-ED* and unlabeled Pcf2 and Pcf3 (*Figure 2—figure supplement 2B*). The $^1$H-$^{15}$N NMR spectrum of this mutant was similar to that of the WT complex, but upon addition of unlabeled histones H3–H4 no major change was observed (*Figure 2—figure supplement 1A*), which strongly suggest an alteration of the histone binding of this mutant.

In summary, we identified critical amino-acids in the ED domain involved in H3–H4 binding. We also showed that addition of histones leads to a conformational change in the *Sp*CAF-1 complex with less disorder in the ED domain and an increased accessibility of the KER domain.

## *Sp*CAF-1 binds dsDNA longer than 40 bp

We next analyzed the DNA binding properties of *Sp*CAF-1. Electrophoretic mobility shift assays (EMSA) were performed with a DNA ladder as substrate in order to determine the minimal DNA size for *Sp*CAF-1 binding. The complex *Sp*CAF-1 showed significant binding for DNAs longer than 40 bp (*Figure 3—figure supplement 1A*). EMSAs with a double-stranded 40 bp DNA fragment showed the formation of a bound complex. When increasing the *Sp*CAF-1 concentration, additional mobility shifts suggest, a cooperative DNA binding (*Figure 3A*). MicroScale thermophoresis (MST) measurements were next performed using an alexa-488 labeled 40 bp dsDNA (*Figure 3B*, *Table 1*). The curves were fitted with a Hill model (*Tso et al., 2018*) with a EC50 value of 0.7±0.1 μM (effective concentration at which a 50% signal is observed) and a cooperativity (Hill coefficient, h) of 2.7±0.2, in line with a cooperative DNA binging of *Sp*CAF-1.

## The KER domain is the main DNA binding region of *Sp*CAF-1

The KER and WHD domains of the CAF-1 large subunit were shown to be involved in DNA binding in *Sc*CAF-1 and *Hs*CAF-1 (*Zhang et al., 2016*; *Sauer et al., 2017*; *Ayoub et al., 2022*; *Gopinathan Nair et al., 2022*). We were thus interested to explore the conservation of these features in Pcf1. We first isolated the KER domain (Pcf1_KER, *Figure 1—figure supplement 1B*), and an extended fragment we called Pcf1_KER-PIP (*Figure 1—figure supplement 1B*), which includes the PIP motif (Q$_{172}$-L-K-L$_{175}$-N-N-F$_{178}$-F$_{179}$). These domains are predicted by AlphaFold2 to form a long helix with

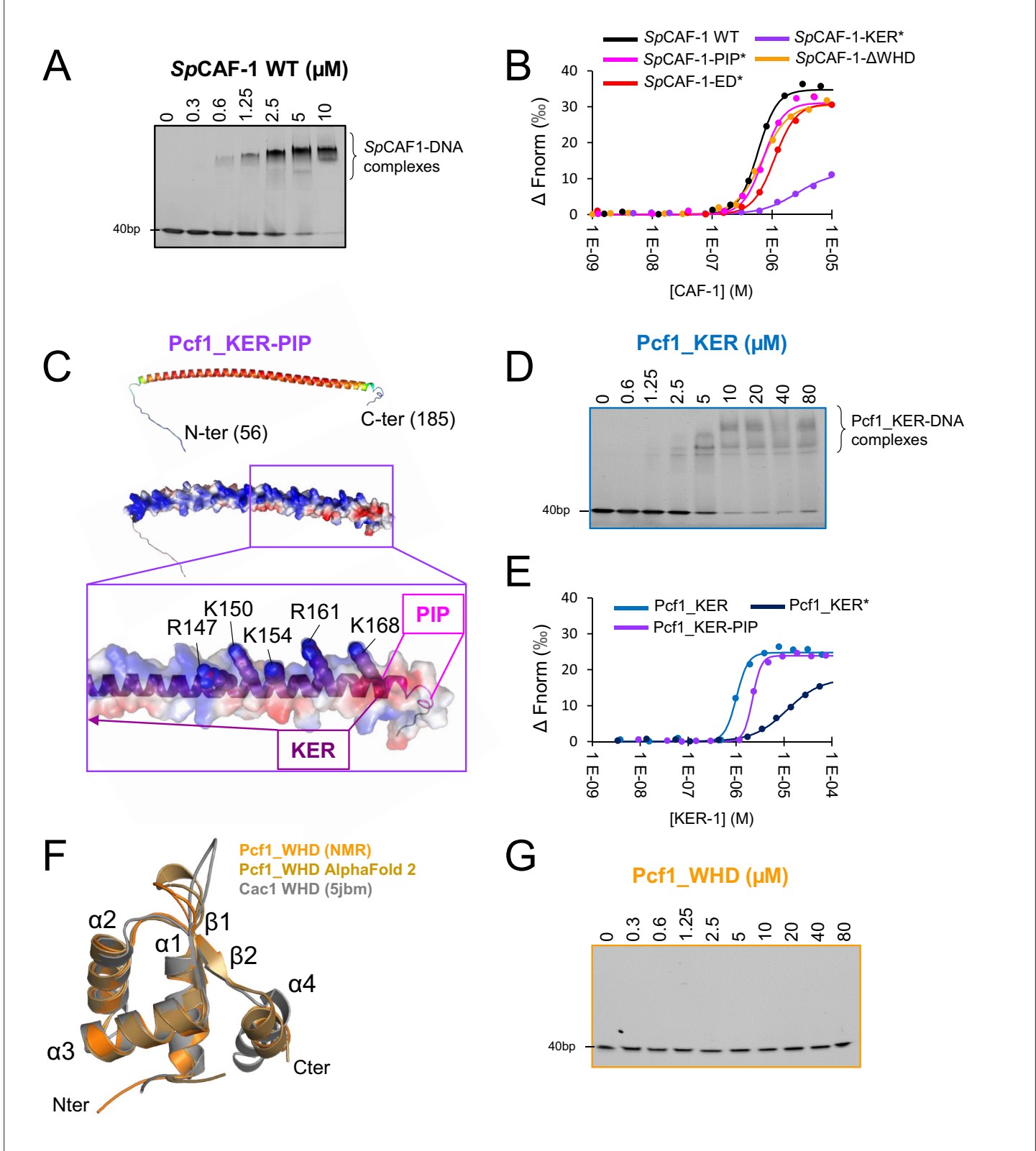

**Figure 3.** Pcf1_KER is the main DNA binding domain of *Sp*CAF-1. (**A**) EMSA with *Sp*CAF-1 and 40 dsDNA (1 μM) revealed with SYBR SAFE staining. (**B**) MicroScale thermophoresis (MST) fitted curves of *Sp*CAF-1 WT and mutants with 40 bp dsDNA. (**C**) Upper panel: Modeled structure of the Pcf1_KER-PIP domain (56-185) rainbow coloured according to the pLDDT of each residue. Red corresponds to pLDDT values of 1 and dark blue of 0. Middle panel same model represented with its electrostatic surface. Lower panel: zoom of the C-terminus of the KER domain and the PIP motif. The five mutated

*Figure 3 continued on next page*

*Figure 3 continued*

residues are labeled and highlighted with spheres. (**D**) EMSA of Pcf1_KER binding with a 40 bp dsDNA (1 μM) revealed with SYBR SAFE staining. (**E**) MST fitted curves of Pcf1_KER constructs and mutants with 40 bp dsDNA. (**F**) Overlay of the calculated model of the WHD domain obtained with the CS-rosetta software (light orange) using NMR assignments of the domain (*Figure 3—figure supplement 4A*), with AlphaFold2 (gold) and the structure of Cac1 WHD from budding yeast (PDB 5jbm, in grey) (*Liu et al., 2016*) (Grey). (**G**) EMSA revealed with SYBR SAFE staining of Pcf1_WHD domain with a 40 dsDNA (1 μM).

The online version of this article includes the following source data and figure supplement(s) for figure 3:

**Source data 1.** Uncropped SDS PAGE gels presented in *Figure 3* and *Figure 3—figure supplements 1–4*.

**Figure supplement 1.** Structural characterization of the Pcf1_KER domain.

**Figure supplement 2.** Binding of WT and mutants Pcf1_KER and *Sp*CAF-1 to DNA analysed by EMSA.

**Figure supplement 3.** Binding of WT and mutants *Sp*CAF-1 to histones H3-H4 analyzed by NMR.

**Figure supplement 4.** Pcf1_WHD domain adopts a WHD fold similar to Sc C-terminal domain of Cac1, but does not bind DNA alone.

partial disorder at both ends and possible extension over the first half of the PIP motif (*Figure 1E*, *Figure 3C*). Using a combination of circular dichroism (CD; *Figure 3—figure supplement 1B–C*), SEC-SAXS (*Figure 1—figure supplement 2C–D*, *Figure 3—figure supplement 1D*) and NMR (*Figure 3—figure supplement 1E–F*), we confirmed that the isolated KER domain of Pcf1 domain forms a straight monomeric helix, partially continuing over the PIP motif. This long helix exhibits a strong bias in amino acid composition and remarkably, almost all basic residues are positioned on the same side of the helix (*Figure 3C*) providing a suitable interface for DNA binding (*Gopinathan Nair et al., 2022*). We performed EMSA using a DNA ladder as substrate and we found that Pcf1_KER domain binds DNA that are longer than 40 bp, as observed with the full *Sp*CAF-1 complex (*Figure 3—figure supplement 2A*). EMSAs with double strand 40 bp DNA fragment showed the presence of multiple bands for Pcf1_KER bound DNA, indicating a possible cooperative DNA binding of this fragment (*Figure 3D*). Affinity measurements by MST led to a EC50 of 1.1±0.2 μM for Pcf1_KER with a cooperativity around 3, consistent with EMSA experiments (*Figure 3E*, *Table 1*). The DNA binding properties of Pcf1_KER-PIP are comparable to that of Pcf1_KER (*Figure 3E*, *Table 1*, *Figure 3—figure supplement 2B*). The EC50 obtained for the isolated Pcf1_KER are also close that of the full *Sp*CAF-1 complex (0.7±0.1 μM) suggesting that the KER domain constitutes the principal DNA binding domain of *Sp*CAF-1.

Based on these results, we designed a mutant called Pcf1_KER* with a charge inversion for five positive residues at the C-terminus of the potential DNA binding face of the KER helix (R147E-K150E-K154E-R161E-K168E; *Figure 3C*, *Figure 1—figure supplement 1B*). The CD analysis of Pcf1_KER* shows this mutant is mainly helical (*Figure 3—figure supplement 2C*). MST quantification confirmed that the mutation of Pcf1_KER* impaired DNA binding by a factor of 10 eventhough residual DNA binding remained (*Figure 3E*, *Figure 3—figure supplement 2D–F*, *Table 1*). The KER* mutation was then introduced in the full complex *Sp*CAF-1-KER* (*Figure 1—figure supplement 1B*, *Figure 2—figure supplement 2B*) and we confirmed by MST and EMSA its lower affinity for dsDNA (*Figure 3B*,

**Table 1.** Experimental affinities of different *Sp*CAF-1 constructs with a 40 bp dsDNA measured by MicroScale thermophoresis (MST) fitted with the Hill model (*Tso et al., 2018*).

| Construct | EC50 (μM) | Hill coeff., h |
|---|---|---|
| Pcf1_KER | 1.1±0.2 | 3.3±0.5 |
| Pcf1_KER* | 12.2±0.7 | 1.5±0.3 |
| Pcf1_KER-PIP | 1.9±0.3 | 5.2±0.9 |
| Pcf1_WHD | Not detected | Not detected |
| *Sp*CAF-1 WT | 0.7±0.1 | 2.7±0.2 |
| *Sp*CAF-1-ΔWHD | 0.7±0.1 | 2.3±0.3 |
| *Sp*CAF-1-KER* | 2.8±0.4 | 1.3±0.3 |
| *Sp*CAF-1-ED* | 1.0±0.1 | 2.3±0.1 |
| *Sp*CAF-1-PIP* | 0.7±0.1 | 2.7±0.3 |

**Table 2.** Interactions parameter with *Sp*PCNA measured by isothermal microcalorimetry (ITC).

| Ligand | Kd (µM) | ΔG (kCal.M⁻¹) | N* | ΔH (kCal.M⁻¹) | -TΔS (kCal.M⁻¹) |
|---|---|---|---|---|---|
| Pcf1_PIP | 7.1±1.3 | −6.9±0.1 | 0.97±0.08 | −2.9±0.2 | −0.39±0.3 |
| Pcf1_PIP* | Not detected | ND | ND | ND | ND |
| Pcf1_KER-PIP | 0.7±0.2 | −8.2±0.2 | 0.64±0.04 | +2.9 ± 0.6 | −11.2±0.8 |
| Pcf1_KER*-PIP | 7.1±1.5 | −6.9±1.2 | 0.7±0.2 | +1.0 ± 0.5 | −7.9±0.7 |
| Pcf1_KER-PIP* | Not detected | ND | ND | ND | ND |
| Cdc27_PIP | 3.5±0.3 | −7.3±0.1 | 0.9±0.1 | −4.8±0.02 | −2.4±0.1 |

*The stoichiometry (N) is calculate as a molar ratio of monomeric PCNA.

*Figure 3—figure supplement 2G*, *Table 1*). Importantly, this mutant also shows a lower binding cooperativity for DNA binding, as estimated by the Hill coefficient value close to 1, compared to values around 3 for the WT and other mutants. In addition, the NMR signals of all IDR for this mutant with or without histones were close to that of the WT (*Figure 3—figure supplement 3A–B*) indicating that the KER* mutation did not impair histone binding.

## The C-terminal of Pcf1 folds as a WHD domain but does not bind DNA

We next isolated the Pcf1_WHD domain (*Figure 1—figure supplement 1B*) and confirmed by NMR and AlphaFold2 that its global fold is similar to *Sc*WHD (*Liu et al., 2016*; *Zhang et al., 2016*; *Figure 3F*, *Figure 3—figure supplement 4A*). Unexpectedly, Pcf1_WHD does not interact with DNA of any size (*Figure 3G*, *Figure 3—figure supplement 4B*). The residues involved in DNA binding in *Sc*WHD, K564, and K568, correspond to S514 and G518 in *Sp*WHD, respectively, leading to a different electrostatic surface, probably not favorable for DNA binding (*Figure 3—figure supplement 4C–D*). To further investigate the role of the WHD domain of *Sp*CAF-1, the WHD domain was deleted in the reconstituted *Sp*CAF-1-ΔWHD complex (*Figure 1—figure supplement 1B*, *Figure 2—figure supplement 2B*) and analyzed by EMSA and NMR. We observed the similar DNA binding property and IDR properties for *Sp*CAF-1-ΔWHD and the WT complex (*Table 1*, *Figure 3B*, *Figure 3—figure supplement 2G*, *Figure 3—figure supplement 3A–B*).

Together our results show that the KER domain constitutes the main DNA binding region of *Sp*CAF-1 and that the WHD domain does not contribute to this binding.

## Crosstalk between DNA and PCNA binding

The PIP motif of Pcf1 was found crucial for *Sp*CAF-1 interaction with PCNA in vivo (*Pietrobon et al., 2014*). Given its proximity with the KER domain, we further investigated potential cross-talks between PCNA and DNA binding. We first measured by isothermal microcalorimetry (ITC) an affinity of 7.1±1.3 µM between *Sp*PCNA and a short PIP motif segment (*Figure 4—figure supplement 1A*, *Table 2*). This affinity is in the same range (twofold less affinity) as a peptide isolated from the replicative polymerase delta from *S. pombe* Cdc27 (*Figure 4—figure supplement 1A*, *Table 2*). In agreement with its consensus sequence, the binding mode of Pcf1_PIP motif to *Sp*PCNA is predicted by AlphaFold to be canonical (*Figure 4—figure supplement 1B*). Consistently, no binding was observed for the Pcf1_PIP* peptide with 4 Alanine mutations, previously designed to disrupt the PIP motif (*Figure 4—figure supplement 1A*, *Table 2*; *Pietrobon et al., 2014*). We next measured the affinity of the longer fragment Pcf1_KER-PIP for *Sp*PCNA and observed an affinity gain of a factor 10 (0.7±1.3 µM; *Figure 4—figure supplement 1C*, *Table 2*), revealing interactions between the KER domain and PCNA. ITC also fits a stoichiometry of ~2 Pcf1_KER-PIP per PCNA trimer, suggesting that, in each PCNA trimer, one monomer remains unbound and potentially accessible for binding to other partners. Pcf1_KER-PIP* did not interact with PCNA confirming the importance of the PIP motif for this association (*Figure 4—figure supplement 1C*, *Table 2*). The KER* mutation impaired the interaction of Pcf1_KER*-PIP with PCNA of a factor 10 reaching the affinity of the short isolated Pcf1 PIP peptide (*Figure 4—figure supplement 1C*, *Table 2*). Collectively these results show that both the PIP motif and the C-terminal part of the KER domain are involved in PCNA binding.

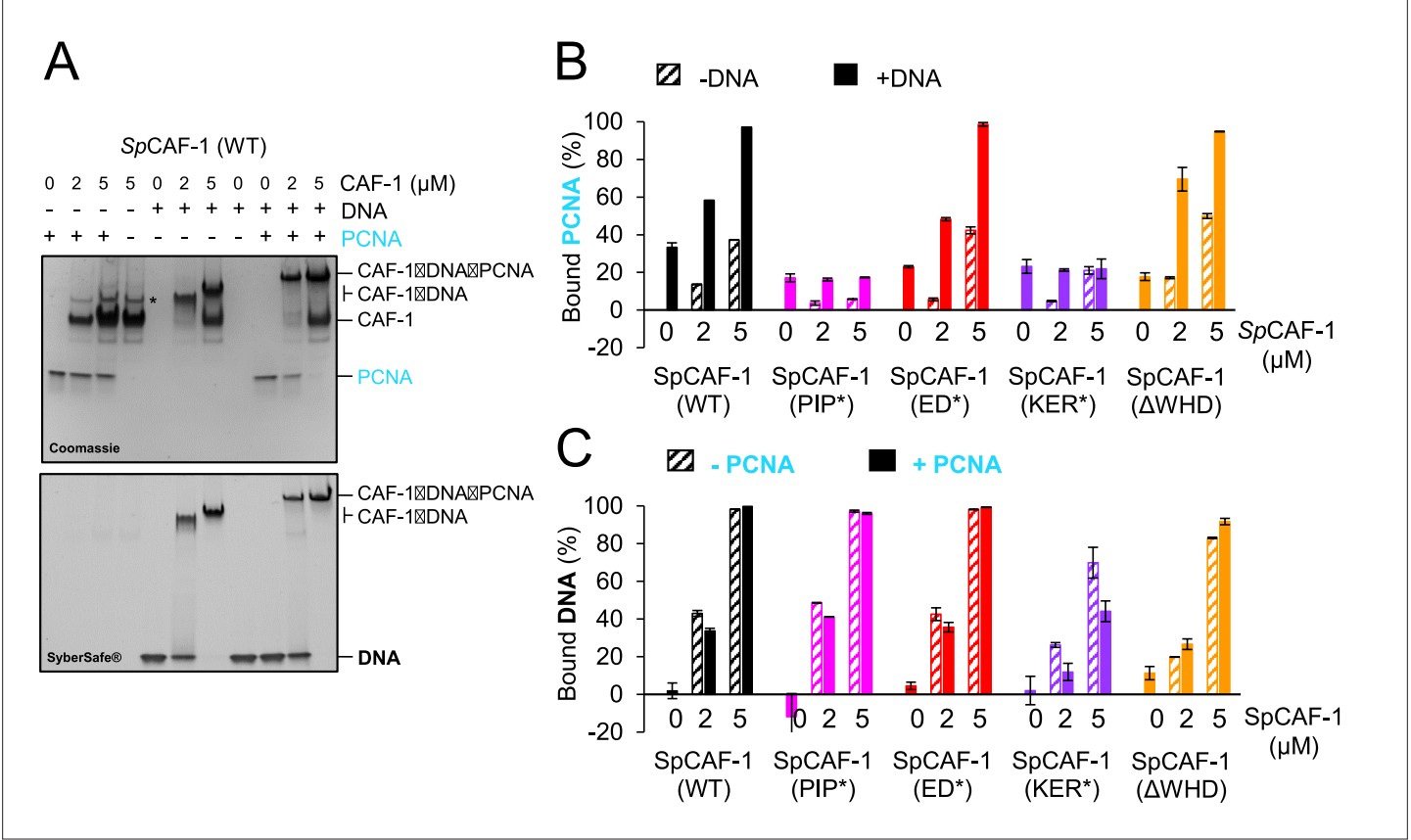

**Figure 4.** The *Sp*CAF-1-KER* mutant is affected for PCNA binding. (**A**) EMSA showing interactions of purified *Sp*CAF-1 (at the indicated concentrations), with or without recombinant *Sp*PCNA (3 µM) in the presence and absence of 40 bp dsDNA (1 µM), revealed with Coomassie blue (upper panel) and with SYBR SAFE staining (lower panel). (**B**) Quantification of bound *Sp*PCNA in the EMSA shown in panel **A** and in *Figure 4—figure supplement 2* for *Sp*CAF-1 and mutants. Values are indicated in % compared to the free PCNA reference (PCNA alone in line 1 in panel **A**) after addition of *Sp*CAF-1 (WT or mutant) at the indicated concentration and in the presence (filled bars) or absence (dashed bars) of 40 bp dsDNA (1 µM). (**C**) Quantification of bound DNA for EMSA shown in panel **A** and in *Figure 4—figure supplement 2* for *Sp*CAF-1 and mutants. Bound DNA in % is compared to the free DNA reference (line 5 in panel **A**) after addition of *Sp*CAF-1 (WT or mutant) at the indicated concentration and in the presence (filled bars) or absence (dashed bars) of *Sp*PCNA (3 µM). All experiments were done in duplicates. Mean values are indicated and error bars shows their standard deviation.

The online version of this article includes the following figure supplement(s) for figure 4:

**Figure supplement 1.** Characterization of the Pcf1 PIP motif.

**Figure supplement 2.** EMSA showing interactions of purified CAF-1 mutated complexes with DNA and PCNA.

To reveal possible crosstalk between CAF-1 binding to PCNA and DNA, we analysed, in the presence or absence of dsDNA, the binding of the full *Sp*CAF-1 complexes (WT *Sp*CAF-1, *Sp*CAF-1-PIP*, *Sp*CAF-1-ED*, *Sp*CAF-1-KER* and *Sp*CAF-1-ΔWHD) with recombinant *Sp*PCNA, using EMSA (*Figure 4A*, *Figure 4—figure supplement 2*). For all combinations tested, we quantified binding by monitoring the disappearance of free PCNA (*Figure 4B*) and free DNA (*Figure 4C*). In this assay, only 20% of free PCNA intensity was lost by addition of DNA (*Figure 4B*), probably because the PCNA trimer can slide along the linear DNA and dissociates during the migration. In the absence of DNA, we observe a small but significant decrease of free PCNA upon addition of WT *Sp*CAF-1, in agreement with the relatively low binding affinity between Pcf1_KER helix and PCNA (*Table 2*). In contrast, in the presence of dsDNA, addition of an excess of WT *Sp*CAF-1 leads to the complete disappearing of the free PCNA band and to a large shift of the band corresponding to *Sp*CAF-1-DNA, corresponding to a larger complex engaging CAF-1, PCNA and DNA (*Figure 4A*). *Sp*CAF-1-ED* and *Sp*CAF-1-ΔWHD show similar binding properties for PCNA and DNA compared to WT CAF-1. In contrast, *Sp*CAF-1-PIP* binds DNA like the WT, but is strongly impaired for PCNA binding alone and in the presence of DNA, while *Sp*CAF-1-KER* is impaired for binding both DNA and *Sp*PCNA. In agreement, the

large shifted band corresponding to a *Sp*CAF-1−PCNA−DNA complex is not observed for these two mutants (*Figure 4—figure supplement 2*).

Altogether, our data show the stabilization of the CAF-1−PCNA interaction by DNA that requires both the KER domain and the PIP motif but not the ED and WHD domain. Conversely, the capacity of CAF-1 to bind PCNA does not impair its interaction with DNA.

## In vitro histone deposition properties of *Sp*CAF-1 mutants

We next examined the ability of the full *Sp*CAF-1 complex reconstituted with the four Pcf1 mutants (*Sp*CAF-1-PIP*, *Sp*CAF-1-ED*, *Sp*CAF-1-KER*, *Sp*CAF-1-ΔWHD) to promote nucleosome assembly mediated by CAF-1 in a complex environment closer to physiological conditions. We used *Xenopus* high-speed egg extract (HSE) that are powerful systems competent for chromatin assembly and effective to exploit depletion/complementation assays (*Ray-Gallet and Almouzni, 2004*). We depleted HSE for the endogenous *Xenopus* CAF-1 largest subunit (xp150) and assessed the capacity of *Sp*CAF-1-PIP*, *Sp*CAF-1-ED*, *Sp*CAF-1-KER*, and *Sp*CAF-1-ΔWHD to complement these xp150-depleted extracts (*Ray-Gallet and Almouzni, 2004*; *Sauer et al., 2017*; *Figure 5—figure supplement 1*). We monitored nucleosome assembly coupled to DNA synthesis using as a template a circular UV-damaged plasmid enabling to analyse by supercoiling assay and nucleotide incorporation simultaneously both repair synthesis and nucleosome formation (*Figure 5*; *Moggs et al., 2000*). We verified that p150-depleted HSE lacked the capacity to promote nucleosome assembly on labeled DNA when compared to mock depleted HSE, and that the recombinant WT *Sp*CAF-1 complex efficiently rescued the loss of xp150 as attested by the detection of supercoiled form I. In contrast, when we complemented the depleted extract with *Sp*CAF-1 mutant complexes *Sp*CAF-1-ED*, *Sp*CAF-1-KER*, *Sp*CAF-1-ΔWHD we did not detect the supercoiled form I. This indicates that these mutants cannot promote nucleosome assembly (*Figure 5*). When we used the *Sp*CAF-1-PIP* mutant, we did not detect supercoiling on labeled DNA at 45 min, yet at 2 hr supercoiling ultimately reached levels achieved using the WT *Sp*CAF-1 (*Figure 5*, bottom, synthesized DNA). Interestingly both for 45 min and 2 hr of assembly *Sp*CAF-1-PIP* mutant yielded more supercoiling than any of the *Sp*CAF-1-ED*, *Sp*CAF-1-KER*, *Sp*CAF-1-ΔWHD mutants. Thus, while mutation in the PIP motif of Pcf1 impaired chromatin assembly at a short time, when more time is given, it allows ultimately to catch up with the wild type. In contrast, none of the *Sp*CAF-1-ED*, *Sp*CAF-1-KER*, *Sp*CAF-1-ΔWHD mutants could catch up, leading to a *Sp*CAF-1 complex deficient for nucleosome assembly even after longer incubation time. Therefore, these data validate the important role of the amino-acids Y340 and W348 within the ED domain in Pcf1 and the importance to preserve the integrity of the KER and WHD domain to ensure a proper *Sp*CAF-1 mediated nucleosome assembly.

Together, these results indicate that the PIP domain provides Pcf1 with the ability to accelerate nucleosome assembly, yet the integrity of the ED, KER and WD domain proved absolutely mandatory for an efficient *Sp*CAF-1 mediated nucleosome assembly.

## Association of *Sp*CAF-1 with histones impacts PCNA interaction in vivo

We next investigated the consequences of the four Pcf1 mutations previously characterized in vitro, on *Sp*CAF-1 function in vivo by introducing the respective mutations at the endogenous *pcf1* gene, a non-essential gene (*Dohke et al., 2008*). Both WT and mutants were FLAG tagged in their N-terminal part. Immunoblot of total cell extract with anti-flag antibody showed that all mutated forms of Pcf1 were expressed to the same level as WT Pcf1 (*Figure 6—figure supplement 1A*).

We first tested PCNA−Pcf1 interaction by co-immunoprecipitation of FLAG-Pcf1 and found that Pcf1-ΔWHD showed a similar PCNA interaction than WT Pcf1 (*Figure 6A–B* and *Figure 6—figure supplement 1B*). No interactions were detected with Pcf1-PIP* and Pcf1-KER*, in line with the requirement of the KER and PIP domains for PCNA binding (*Figure 4*, *Table 2*). Surprisingly, we found that Pcf1-ED* binds eight times more to PCNA than the WT Pcf1 (*Figure 6A–B*) although the corresponding *Sp*CAF-1-ED* bound PCNA with or without DNA in vitro, similarly to WT (*Figure 4*, *Figure 4—figure supplement 2*). This suggests an interplay in vivo between the binding of CAF-1 to PCNA and its capacity to bind histones.

We next analyzed the ability of CAF-1 mutated forms to form foci during S-phase (*Pietrobon et al., 2014*). Since previously reported GFP-tagged forms of Pcf1 are not fully functional, we made use of cells expressing Pcf2-GFP, a functional tagged form (*Hardy et al., 2019*; *Figure 6C*). As expected,

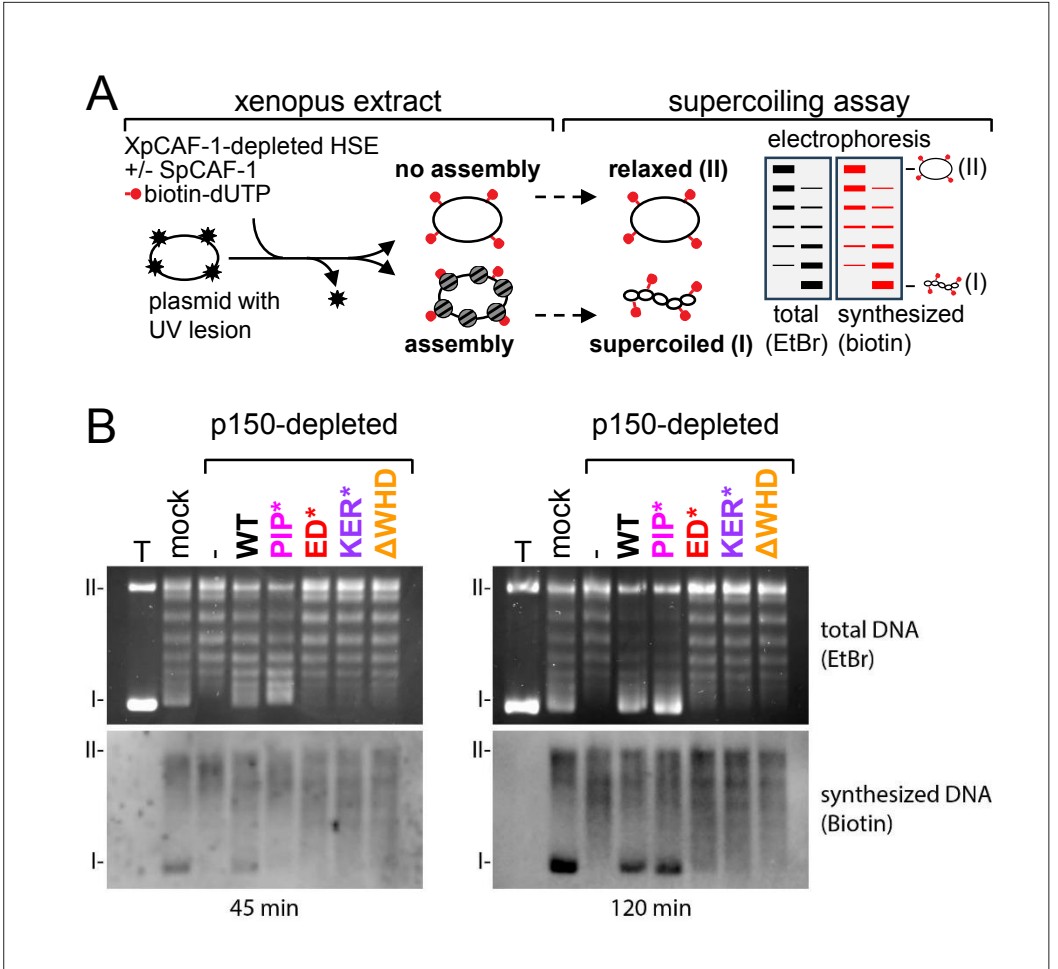

**Figure 5.** Efficient nucleosome assembly by *Sp*CAF-1 in vitro requires interactions with H3–H4, DNA and PCNA, and the C-terminal WHD domain. (**A**) Experimental scheme depicting the nucleosome assembly assay to monitor the efficiency of the *Sp*CAF-1 complex. A plasmid with UV lesions (black star) when incubated in *Xenopus* HSE extracts undergoes DNA repair synthesis and the appearance of supercoiling is indicative of nucleosome assembly. In Extracts depleted from CAF-1, this nucleosome assembly coupled to DNA repair synthesis is stimulated by CAF-1 addition. The supercoiling assay separates by gel electrophoresis the relaxed plasmids (form II) not assembled from assembled plasmids, fully supercoiled (form I). DNA synthesis is monitored by biotin detection of biotin-dUTP incorporation (red). (**B**) Gel electrophoresis after 45 (left) and 120 (right) min incubation to monitor chromatin assembly in control mock and *Xenopus* p150-depleted HSE. Total DNA visualized by EtBr staining (top) and synthesized DNA visualized by biotin detection (bottom) are shown. The *Xenopus* p150-depleted HSE is either mock complemented (-) or complemented using *Sp*CAF-1 complex composed of wild type Pcf1(WT) or mutants Pcf1-PIP*, Pcf1-ED*, Pcf1-KER*-, or Pcf1-ΔWHD- as indicated. T: pBS plasmid incubated without extract run in parallel serves as a migration control to locate supercoiled DNA. The position of relaxed (II) and supercoiled (I) DNA are indicated.

The online version of this article includes the following figure supplement(s) for figure 5:

**Figure supplement 1.** Western blot analysis of mock- and p150-depleted HSE.

Pcf2-GFP formed discrete foci during the bulk of S-phase (in bi-nucleated cells with septum) but not during G2 phase (mono-nucleated cells) in a Pcf1-dependent manner (*Figure 6C–D* and *Figure 6—figure supplement 1C*). The *pcf1-ΔWHD* mutation behaved like the WT in this assay, whereas S-phase Pcf2 foci were undetectable when Pcf1−PCNA interaction is impaired in *pcf1-KER** and *pcf1-PIP** (*Figure 6C–D*). Interestingly, Pcf2-GFP foci were more frequent in all cell cycle phases in *pcf1-ED** mutated cells compared to WT. Simultaneous acquisition of GFP fluorescence in living WT and mutated *pcf1* cells revealed that Pcf2-GFP foci were more abundant and brighter in *pcf1-ED** cells compared to WT (*Figure 6C*), suggesting a higher concentration of CAF-1 within replication factories.

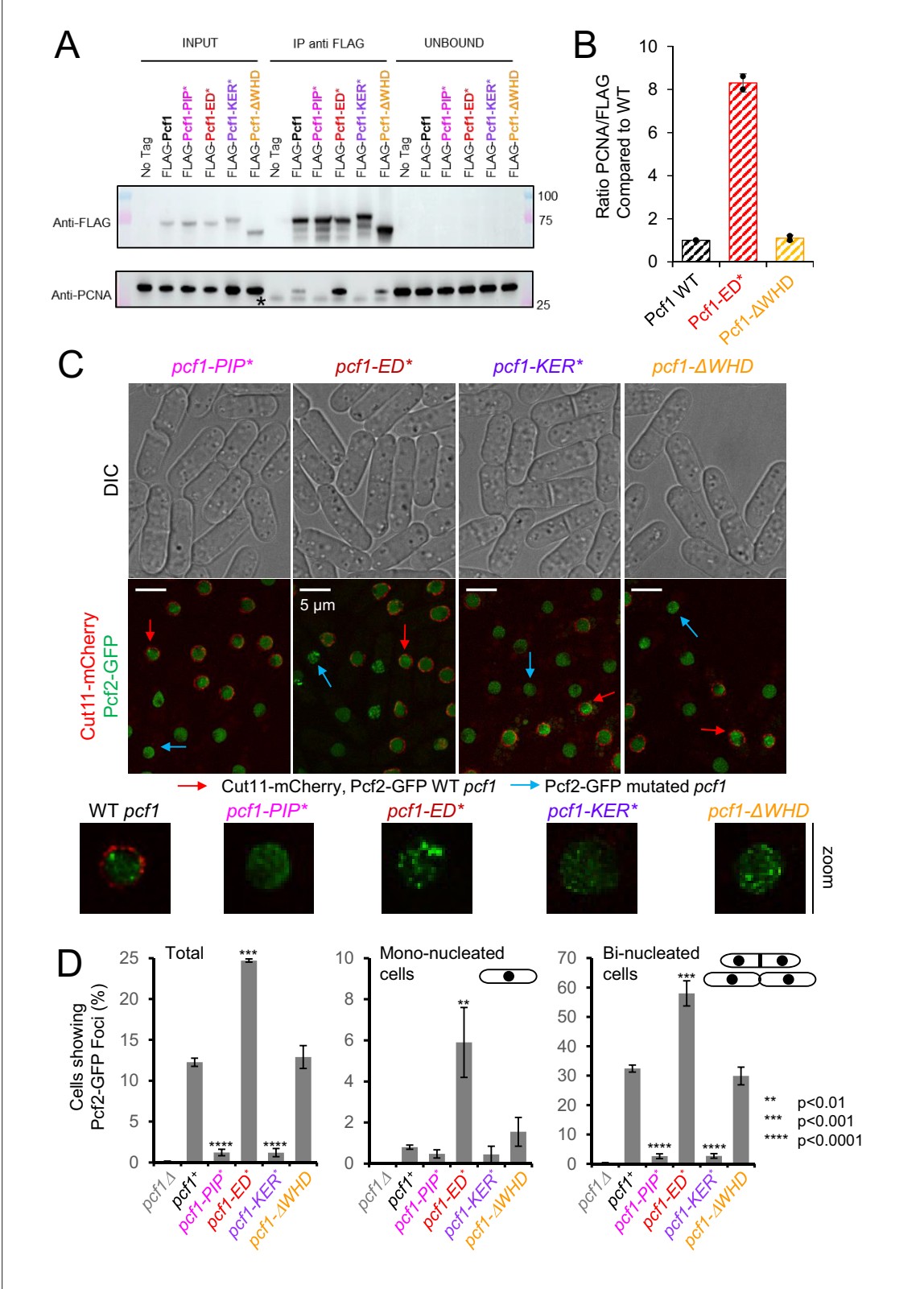

**Figure 6.** Association of CAF-1 with histone modulates PCNA interaction in vivo and foci formation. (**A**) Anti-FLAG Pulldown to address PCNA−CAF-1 interaction in vivo in indicated strains. (**B**) Quantification of bound PCNA from (**A**). (**C**) Simultaneous acquisition of Pcf2-GFP in WT (red arrow) and *pcf1* mutated strains (blue arrow). Strains were grown separately and equally mixed before to process them for cell imaging. WT *pcf1* cells expressed the fusion Cut11-mCherry, a component of the nuclear pore complex, leading to a red labelling of the nuclear periphery. Under same illumination and

*Figure 6 continued on next page*

*Figure 6 continued*

acquisition conditions, Pcf2-GFP foci are detected in WT and *pcf1-ΔWHD* cells, but not in *pcf1-PIP** and *pcf1-KER** cells. In contrast, Pcf2-GFP foci are brighter and more abundant in *pcf1-ED** cells. (**D**) Quantification of cells showing Pcf2-GFP foci, according to cell morphology in indicated strains. Mono-nucleated cells mark G2-phase cells and bi-nucleated cells with septum mark S-phase cells. Values are means of at least three independent experiments ± standard error of the mean (sem). At least 1000 nuclei were analysed per strain. p Values are indicated with stars and were calculated using the student test.

The online version of this article includes the following source data and figure supplement(s) for figure 6:

**Source data 1.** Uncropped blots presented in *Figure 6A*.

**Figure supplement 1.** Association of CAF-1 with histone is coupled to PCNA interaction in vivo.

In conclusion, CAF-1 foci in S-phase correlate with its association with PCNA in vivo, possibly modulated by the histone binding.

## The WHD domain specifies CAF-1 function in distinct cellular processes

In *S. pombe*, CAF-1 is involved in the replication-coupled maintenance of heterochromatin (*Dohke et al., 2008*). We employed a strain in which *ura4⁺* is inserted at the peri-centromeric heterochromatin of the chromosome I (*Figure 7A*, top panel). The expression of *ura4* is repressed by the surrounding heterochromatin resulting in a poor growth on uracil-depleted media and resistance to 5-fluoro-orotic acid (5FOA) (*Figure 7A*, bottom panel). As previously reported, the deletion of *pcf1* resulted in a better cell growth on uracil-depleted media compared to WT cells, showing that the heterochromatin is not properly maintained, leading to the derepression of *ura4⁺*. All mutants, excepted *pcf1-ΔWHD*, exhibited defects in *ura4* silencing, similar to the one observed in the null mutant. This shows that the inability to interact with histone, PCNA and DNA results in a complete lack of CAF-1 function in maintaining heterochromatin. Interestingly, the WHD domain, while required for chromatin assembly in vitro (*Figure 5*), is dispensable for the maintenance of heterochromatin. We thus investigated further the role of this domain.

We analyzed the accumulation of Rad52-GFP foci as a readout of global accumulation of DNA damage (*Figure 7B–C*). The deletion of *pcf1* led to a modest but significant increase in the frequency of cells showing Rad52-GFP foci. A similar effect was observed in *pcf1-ED** mutated cells, while the presence of a CAF-1 complex unable to interact with PCNA resulted in a greater increase (in *pcf1-PIP** and *pcf1-KER** mutants). In contrast, no significant increase was observed in *pcf1-ΔWHD* cells. Thus, both CAF-1 interaction with histone and PCNA prevent the accumulation of DNA damage, but histone deposition is not absolutely required.

The deletion of *pcf1* is synthetic lethal with the deletion of *hip1*, the gene encoding one subunit of the fission yeast HIRA complex (*Hardy et al., 2019*), indicating that in the absence of replication-coupled histone deposition by CAF-1, cell viability relies on H3–H4 deposition by HIRA as suggested in human and *S. cerevisiae* (*Kaufman et al., 1998*; *Krawitz et al., 2002*; *Ray-Gallet et al., 2011*). We found that *pcf1-KER**, *pcf1-ΔWHD* or *pcf1-PIP** are co-lethal with *hip1* deletion (*Figure 7D*). Spores harboring *pcf1-ED** were viable when combined with *hip1* deletion, but exhibited a severe growth defect (*Figure 7D* and *Figure 7—figure supplement 1*), suggesting that CAF1-ED* complexes can still perform some histone deposition in vivo. These genetic interactions indicate that binding of CAF-1 to PCNA, DNA and histones are critical determinants for its function in vivo, as well as the WHD C-terminal domain.

## Discussion

In the present work, we provide a comprehensive and dynamic view of key structural features of the histone chaperone CAF-1 from *S. pombe*. Despite the low sequence conservation between orthologues of the large subunit of CAF-1 (*Figure 1—figure supplement 1A*), Pcf1 from *Sp*CAF-1 mediates the heterotrimer complex that binds dimeric histones H3–H4, as does *Sc*CAF-1 (*Kaufman et al., 1995*; *Liu et al., 2012*; *Mattiroli et al., 2017b*, *Sauer et al., 2017*). Using AlphaFold2, we propose a structural model of the *Sp*CAF-1, fully supported by our experimental data (*Figure 1*, *Figure 1—figure supplements 2–4*). This structure defines the 2BD and 3BD regions in Pcf1 as involved in the binding of Pcf2 and Pcf3, respectively. This matches remarkably the corresponding segments identified by HDX in *Sc*CAF-1 (*Mattiroli et al., 2017a*). In line with previous observations in *Hs*CAF-1

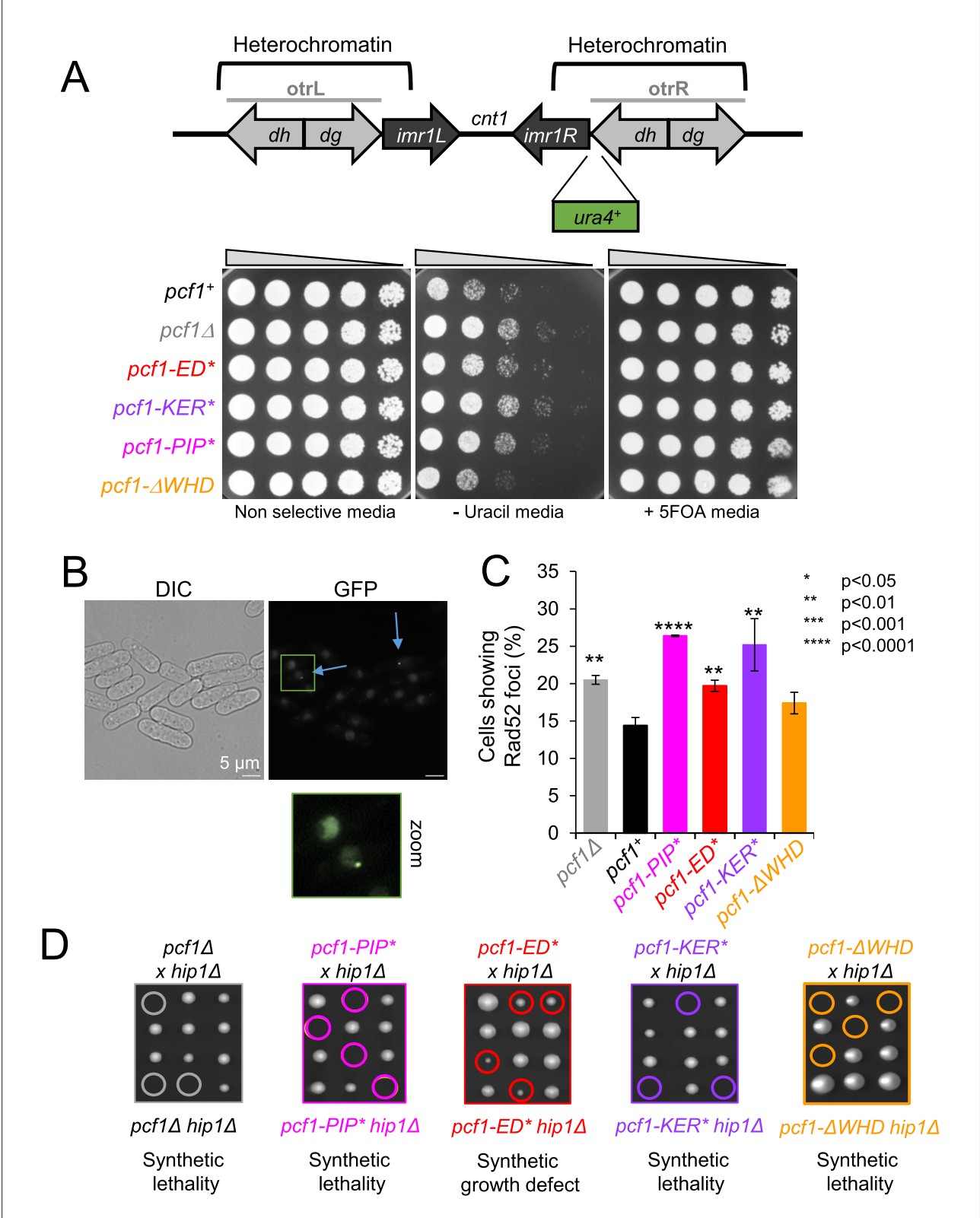

**Figure 7.** The WHD domain of *Sp*CAF-1 specifies CAF-1 function. (**A**) Top panel: Schematic representation of the silencing assay used. Otr: outer repeats; imr: inner repeats; cnt1: central core of the centromere 1. Bottom panel: Serial fivefold dilution of indicated strains on indicated media. (**B**) Example of Rad52-GFP foci in WT cells. Blue arrows indicate Rad52 foci-positive cells. (**C**) Quantification of Rad52-GFP foci in indicated strains. Values are means of at least three independent experiments ±sem. p Values are indicated as stars and were calculated with the student test. At least 1000

*Figure 7 continued on next page*

*Figure 7 continued*

nuclei were analyzed per strain. (**D**) Co-lethality assay. Tetrad dissections of cells deleted for *hip1* (*hip1Δ*) crossed with cells deleted for *pcf1* (*pcf1Δ*) (grey) or harboring *pcf1-PIP\** (magenta), *pcf1-ED\** (red), *pcf1-KER\** (purple), or *pcf1-ΔWHD\** (orange). Spores with double mutations are surrounded and were deducted from the analysis of viable spores from each tetrad (see Materials and methods). At least 18 tetrads were analysed per cross.

The online version of this article includes the following figure supplement(s) for figure 7:

**Figure supplement 1.** the *pcf1-ED\** mutation confers a synthetic growth defect when combined with *hip1Δ*.

---

(*Kaufman et al., 1995*) and *Sc*CAF-1 (*Liu et al., 2016*; *Mattiroli et al., 2017b, Mattiroli et al., 2017a*), the ED domain of *Sp*CAF-1 is crucial for histone binding (*Figure 2*). Mainly disordered in the free chaperone, we show that this domain folds upon histone binding, promoting a conformational change with increased accessibility of the KER domain (*Figure 1—figure supplement 5*). *Sp*CAF-1 binds dsDNA longer than 40 bp in the micromolar affinity range (*Figure 3*, *Table 1*, *Figure 3—figure supplements 1–2*) through the KER domain forming a long monomeric helix with a positively charged face. Interestingly, the helix length roughly corresponds to the size of 40 bp dsDNA, suggesting that it could lie on DNA and act as a DNA ruler to sense free DNA for histone deposition (*Sauer et al., 2017*; *Gopinathan Nair et al., 2022*; *Rosas et al., 2023*). Together, our findings highlight the conservation of CAF-1 properties in histone deposition mechanism in vitro, and thus unifies the current model (*Sauer et al., 2018*).

This work revealed strong interdependency between histone deposition by CAF-1 and its association with PCNA. The PIP* mutation did not compromise DNA binding of *Sp*CAF-1 in vitro (*Figure 4—figure supplement 2*). Conversely, upon interaction with DNA, *Sp*CAF-1 interacted tighter with PCNA (*Figure 4*), consistently with a recent study in budding yeast (*Rouillon et al., 2023*). We show that *Sp*CAF-1-PIP* is still able to assemble histones in vitro, although slower than WT-*Sp*CAF-1 (*Figure 5*). In contrast, in vivo, *pcf1-PIP\** phenocopy the deletion of *pcf1* (*Figures 6–7*). From these results, we conclude that the binding of *Sp*CAF-1 to PCNA through the PIP motif is required for *Sp*CAF-1 functions in vivo, by allowing its recruitment and efficient histone deposition at DNA synthesis sites.

*Sp*CAF-1-ED* showed a stronger interaction with PCNA than WT-*Sp*CAF-1 in vivo, and formed more abundant and intense foci during S-phase. (*Figure 6*). This default may not result from a direct competition between PCNA and histones for CAF-1 association since *Sp*CAF-1-ED* and WT-*Sp*CAF-1 show similar interaction with DNA and PCNA in vitro (*Figure 4*, *Figure 4—figure supplement 2*). In human cells lacking new histones, PCNA accumulates on newly synthetized DNA, and PCNA unloading has recently linked to histone deposition in budding yeast (*Mejlvang et al., 2014*; *Janke et al., 2018*; *Thakar et al., 2020*). We propose that the accumulation of CAF-1 at replication foci in the ED* mutant may reflect PCNA recycling defects. This cannot be attributed to the inability of *Sp*CAF-1-ED* to deposit histones otherwise we should have observed the same accumulation for *Sp*CAF-1-ΔWHD also defective for histone deposition. The ED* mutation could rather interfere with other interactions, or with post-translational modifications (PTMs) contributing to recycle PCNA.

Deletion of the WHD domain allowed separating *Sp*CAF-1 functions in chromatin assembly, heterochromatin maintenance and the prevention of DNA damage. Unlike *Sc*CAF-1 (*Liu et al., 2016*; *Zhang et al., 2016*; *Mattiroli et al., 2017b, Mattiroli et al., 2017a*) and *Hs*CAF-1[28], Pcf1_WHD did not bind DNA nor the ED domain (which remains fully disordered) in the free chaperone. Nevertheless, on the NMR spectra of the free and histone bound *Sp*CAF-1([15]N-Pcf1), the resonances of the isolated WHD domain are not present (*Figure 1D*, *Figure 3—figure supplement 4A*), in agreement with a restricted movement of this domain that could likely interacts with other folded parts of the complex. In vitro, we found no impact of the WHD deletion on CAF-1 interaction with DNA, histones or PCNA, but the *Sp*CAF-1-ΔWHD was deficient for histone deposition. Thus, the synthetic lethality of this mutant with *hip1* most likely reflects a replication-coupled assembly defect. Unexpectedly, this defect does not cause a problem of heterochromatin maintenance or damage accumulation, showing that the multiple functions of CAF-1 in replication-dependent nucleosome assembly, genome stability and heterochromatin maintenance can be uncoupled thanks to the WHD domain contributing to specify CAF-1 functions. Further investigations will be necessary to understand the role of this domain.

We reveal that disorder is a fundamental feature of Pcf1 supporting its molecular functions. First, the ED domain is disordered in the FL complex and folds upon histone binding. Second, four IDRs demarcate specific domains within Pcf1. We believe that these unfolded regions provide unique 'plasticity' properties to Pcf1 allowing these domains to bind concomitantly their multiple specific partners

(Pcf1, Pcf3, PCNA, DNA, and histones). We also reveal that although these domains individually bind their specific partners, there is an important crosstalk between them as exemplified by the fact that DNA stabilizes the CAF-1–PCNA interaction. Such plasticity and cross-talks provided by structurally disordered domains might be key for the multivalent CAF-1 functions. Human CAF-1 has been reported to form nuclear bodies with liquid-liquid phase separation properties to maintain HIV latency (*Ma et al., 2021*). This raises the question of a potential role of the disordered domains of Pcf1, together with other replisome factor harboring such disordered regions (*Bedina, 2013*), in promoting phase separation of replication factories, if such phenomenon happens in vivo. Further studies will be needed to tackle these questions.

## Materials and methods
### Plasmid preparation for recombinant protein production
The cDNA sequence of WT Pcf1 (codon optimized for *E. coli* expression) was synthetized and inserted into the pCM153 plasmid to obtain the recombinant MBP–6His-TEV cleavage site-Pcf1 protein (named MBP-Pcf1 below). The cDNA sequence of WT Pcf2 and WT Pcf3 (codon optimized for insect cells expression) were synthetized and introduced into a pKL plasmid for protein expression in insect cells (MultiBac approach *Berger et al., 2004*) with either a C-terminal (for Pcf2) or a N-terminal (for Pcf3) 6His tag with a TEV cleavage site between the protein and the His tag. Pcf1_ED (325-396) and Pcf1_WHD (471-544) were sub cloned in frame into pET28A-B18R plasmid for expression with a N-terminal 6His-SUMO tag. Pcf1_KER (56-170) and Pcf1-KER-PIP (56-185) were inserted in frame into pCM153 plasmid *Miele et al., 2022* for expression with a N-terminal 6His-MBP-TEV tag. The cDNA sequence if *S. pombe* histones H3–H4 (codon optimized for *E. coli* expression) were introduced in the 6His-dAsf1 from the pET28 plasmid (generous gift from R.N. Dutnall) in place of histones *Dm*H3–H4 (*Anderson et al., 2010*). With this vector, histones H3–H4 are coexpressed with the chaperone ASF1, leading to soluble untagged free-histones, and ASF1-bound histones. The cDNA of *Sp*PCNA (codon optimized for *E. coli* expression) was synthetized and inserted into the pET28A-B18R plasmid for expression with an N-terminal 6His-SUMO. Pcf1 mutants were generated by PCR. All plasmids for recombinant protein expression were constructed by GenScript.

### Recombinant protein production
Pcf1 was overexpressed in *E. coli*. After fresh transformation of *E. coli* BL21 (DE3) Star cells (Thermo Fisher Scientific), cells were grown in an auto-induction rich medium Terrific Broth (12 g/L tryptone, 24 g/L yeast extract) containing 50 µg/mL of Kanamycin for 30 hr at 20 °C, under agitation. *Sp*Histones H3–H4, *Sp*PCNA and the all domains of Pcf1 were overexpressed in *E. coli*. The plasmid for expressing the desired protein was freshly transformed in *E. coli* strain BL21 DE3 STAR (Thermo Fisher Scientific). Cells were grown at 37 °C in LB medium containing 50 µg/mL of Kanamycin until OD reached 0.7 and recombinant protein expression was induced for 16 hr at 20 °C under agitation by adding 1 mM isopropy β-D-1-thiogalactopyranoside IPTG, or cells were grown 30 hr at 20 °C in a ZY auto-inducible medium. For $^{15}$N or $^{13}$C uniformly labeled proteins, the expression was made in minimal media with 0.5 g/L of $^{15}$NH$_4$Cl and/or 2 g/L of $^{13}$C-glucose. Pcf2 and Pcf3 were produced in insect cells. Sf9 Insect cells were infected with an MOI of 5*10–3 virus/cell and incubated for 5 days at 27 °C at 130 rpm. After centrifugation, cell pellets stored at –70 °C until further use.

### Protein purifications
Purification of Pcf1
Cells were pelleted by centrifugation and resuspended in the lysis buffer LB1 for 30 min (50 mM Tris-HCl pH 8, 500 mM NaCl, 5% glycerol, 0.1% Triton X-100, 2 mM DTT, 5 mM MgCl2, 0.5 mM PMSF, 1 X cOmplete EDTA-free Protease Inhibitor Cocktail, 1.2 mg/mL lysozyme and 70 U/mL of benzonase). Cells were lysed by sonication at 4 °C, the lysate was clarified by centrifugation at 5 °C at 18500 rpm for 30 min and loaded onto gravity flow amylose resin (NEB) previously equilibrated with buffer WB1_1 (50 mM Tris-HCl pH 8, 500 mM NaCl, 2 mM DTT). After loading the cell lysate onto the resin, the resin was washed with 5 column volumes of buffer WB1_1 to ensure complete passage of the cell lysate through the resin. Then, the resin was further washed with 10 column volumes of buffer WB1_2 (50 mM Tris-HCl pH 8, 1000 mM NaCl, 2 mM DTT) to remove non-specific

binding, before re-equilibration with 10 column volumes of buffer WB1_1. MBP-Pcf1 was eluted with 10 column volumes of buffer EB1 (50 mM Tris-HCl pH 8, 500 mM NaCl, 0.5 mM TCEP, 10 mM maltose and 1 X cOmplete EDTA-free Protease Inhibitor Cocktail). After addition of 1 mM MgCl, the eluate containing MBP-Pcf1 was incubated 16 hr at 5 °C with TEV protease (added with a ratio 1/20 in mass). The eluate was then concentrated to 300 µL (with Amicon Ultra-15 30 kDa filter concentrators), 2000 U of benzonase were added and incubated for 2 hr. The concentrated eluate was injected into a column Superose 6 increase 10/300 GL (Cytiva) previously equilibrated with the final buffer FB1 (50 mM Tris-HCl pH 8, 500 mM NaCl, 1 mM DTT). The Pcf1-containing fractions were pooled. 1 X cOmplete EDTA-free Protease Inhibitor Cocktail, 0.5 mM TCEP and 30% glycerol was added and samples were snap-frozen and stored at –70 °C.

## Purification of Pcf2

Cells pellets were resuspended into lysis buffer LB2 (50 mM Tris-HCl pH 8, 500 mM NaCl, 5% glycerol, 0.1% Triton X-100, 10 mM imidazole, 0.5 mM PMSF, cOmplete EDTA-free Protease Inhibitor Cocktail and 70 U/mL of benzonase) and sonicated at 4 °C. Lysates were clarified by centrifugation at 5 °C at 18500 rpm for 30 min and loaded to gravity flow Ni-NTA agarose resin (QIAGEN) previously equilibrated with wash buffer WB2_1 (50 mM Tris-HCl pH 8, 500 mM NaCl, 10 mM imidazole). Resin was then washed with 10 column volumes of wash buffer WB2_1 followed by 10 column volumes of wash buffer WB2_2 (50 mM Tris-HCl pH 8, 1 M NaCl, 10 mM imidazole). Pcf2-6His was eluted with 5 column volumes of the elution buffer EB2 (50 mM Tris-HCl pH 8, 500 mM NaCl, 250 mM imidazole, 0.5 mM TCEP, and 1 X cOmplete EDTA-free Protease Inhibitor Cocktail). The eluate was then concentrated to 300 µL (with Amicon Ultra-15 30 kDa filter concentrators). 1 mM DTT, 1 mM MgCl and ≈2000 U of benzonase were added to the concentrated eluate and the sample was incubated 2 hr at 4 °C and injected into a Superdex 200 increase 10/300 (Cytiva) previously equilibrated with the final buffer FB2 (50 mM Tris-HCl pH 8, 500 mM NaCl and 1 mM DTT). Pcf2-6His containing fractions were pooled and directly used for CAF-1 reconstitution or stored at –70 °C with cOmplete EDTA-free Protease Inhibitor Cocktail, 0.5 mM TCEP and 30% glycerol.

## Purification of Pcf3

Cells pellets were resuspended into lysis buffer LB3 (50 mM Tris-HCl pH 8, 200 mM NaCl, 5% glycerol, 0.1% Triton X-100, 10 mM imidazole, 0.5 mM PMSF, cOmplete EDTA-free Protease Inhibitor Cocktail and 70 U/mL of benzonase) and sonicated at 4 °C. Lysate was clarified by centrifugation at 5 °C at 18500 rpm for 30 min and loaded to gravity flow Ni-NTA agarose resin (QIAGEN) previously equilibrated with wash buffer WB3_1 (50 mM Tris-HCl pH 8, 200 mM NaCl, 10 mM imidazole). Resin was then washed with 5 column volumes of wash buffer WB3_1 and 10 column volumes of wash buffer WB3_2 (50 mM Tris-HCl pH 8, 200 mM NaCl, 30 mM imidazole). his-Pcf3 was then eluted with EB3 (50 mM Tris-HCl pH 8, 200 mM NaCl, 250 mM imidazole, 1 mM DTT, 1 X cOmplete EDTA-free Protease Inhibitor Cocktail). After addition of 1 mM MgCl2, 6His-TEV protease (with a ratio 1/10 in mass), ≈2000 U of benzonase the eluate was dialysed o/n at 5 °C in the final buffer FB3 (50 mM Tris-HCl pH 8, 200 mM NaCl and 1 mM DTT). Because of their similar size, Pcf3 and 6His-TEV protease cannot be completely separated by size-exclusion chromatography. Therefore, to remove the 6His-TEV protease and uncleaved His-Pcf3, 30 mM of imidazole was added to the dialysate, which was then loaded to gravity flow Ni-NTA agarose resin (QIAGEN) previously equilibrated with wash buffer WB3_2. The flow through was concentrated to 300 µL (with Amicon Ultra-15 30 kDa filter concentrators) and injected into a Superdex 200 increase 10/300 (Cytiva) previously equilibrated with the final buffer FB3. Pcf3-containing fractions were pooled and stored at –70 °C after adding 1 X cOmplete EDTA-free Protease Inhibitor Cocktail, 0.5 mM TCEP and 30% glycerol.

## Reconstitution of CAF-1 complexes

CAF-1 complexes were formed by mixing the isolated proteins Pcf1 (WT or mutant), Pcf2-6His and Pcf3 previously purified as described above. Isolated Pcf2-6His and Pcf3 were added in small excess compared to Pcf1. Tris 50 mM pH 8 was added to the Pcf1/Pcf2-his/Pcf3 mix to reach a final NaCl concentration of 150 mM. After addition of 1 mM MgCl2, 1 X cOmplete EDTA-free Protease Inhibitor Cocktail, 6His-TEV protease (with a ratio 1/10 in mass) and ≈2000 U of benzonase, the mixture was incubated over night at 4 °C and applied on a HiTrap heparin FF column (Cytiva) previously

equilibrated with EB4_1 (50 mM Tris-HCl pH 8, 100 mM NaCl). A gradient was applied with the high salt buffer EB4_2 (50 mM Tris-HCl pH 8, 1 M NaCl). Fractions containing the full $Sp$CAF-1 were pooled, concentrated to 300 µL (with Amicon Ultra-15 30 kDa filter concentrators) and injected into a Superdex 200 increase 10/300 (Cytiva) previously equilibrated with the final buffer FB4_1 (50 mM Tris-HCl pH 8, 150 mM NaCl and 1 mM DTT). The $Sp$CAF-1-containing fractions corresponding to the 4b peak (9.7 mL, *Figure 1—figure supplement 2A*) were pooled and directly used for structural analyses and DNA/PCNA interactions including MST or EMSA. Aliquots were flash-frozen and stored at –70 °C with 1 X cOmplete EDTA-free Protease Inhibitor Cocktail, 0.5 mM TCEP and 30% or 50% glycerol for in vitro nucleosome assembly assays (*Figure 5*).

The $Sp$CAF-1($^{15}$N-$^{13}$C-Pcf1) and $Sp$CAF-1($^{15}$N-Pcf1) was reconstituted by co-lysing the pellets of $^{15}$N-$^{13}$C-MBP-Pcf1 or $^{15}$N-MBP-Pcf1 (WT or mutants), Pcf2-6His and his-Pcf3. Cell pellets from Pcf2-6His and Pcf3-his were added in excess compared to labeled MBP-Pcf1, based on the yield previously obtained for the isolated proteins. The pellets were resuspended and mixed in the lysis buffer LB4 (50 mM Tris-HCl pH 8, 150 mM NaCl, 5% glycerol, 0.1% Triton X-100, 10 mM imidazole, 0.5 mM PMSF, cOmplete EDTA-free Protease Inhibitor Cocktail and 70 U/mL of benzonase), sonicated and centrifuged as described before. The clarified lysate was applied to gravity flow Ni-NTA agarose resin (QIAGEN) previously equilibrated with wash buffer WB4_1 (50 mM Tris-HCl pH 8, 150 mM NaCl, 10 mM imidazole). Beads were washed with 5 column volume of WB4_1 buffer, followed by 10 column volumes of WB4_2 (50 mM Tris-HCl pH 8, 1 M NaCl, 10 mM imidazole). Elution was performed with EB4 (50 mM Tris-HCl pH 8, 150 mM NaCl, 250 mM imidazole, 1 X cOmplete EDTA-free Protease Inhibitor Cocktail) and applied to an anion exchange column HiTrap Q FF (Cytiva) previously equilibrated with buffer EB4_1. A gradient was applied with the high salt buffer EB4_2. The tagged CAF-1-containing fractions were pooled, and dialyzed overnight against buffer 4 DB4 (Tris 50 mM pH 8, 150 mM NaCl, 1 mM DTT) after addition of 1 mM DTT, 1 mM MgCl2, 1 X cOmplete EDTA-free Protease Inhibitor Cocktail, 6His-TEV protease (with a ratio 1/10 in mass) and ≈2000 U of benzonase. The mixture was applied on a HiTrap heparin FF column (Cytiva) using the same buffers (EB4_1 and EB4_2). The $Sp$CAF-1($^{15}$N-$^{13}$C/$^{15}$N-Pcf1) containing fractions were concentrated to 300 µL (with Amicon Ultra-15 30 kDa filter concentrators) and injected into a Superdex 200 increase 10/300 (Cytiva) previously equilibrated with buffer FB4_2 (10 mM Tris-HCl, 50 mM HEPES pH 7, 300 mM NaCl and 0.5 mM TCEP). The $Sp$CAF-1($^{15}$N-$^{13}$C-Pcf1)-containing fractions corresponding to the 4b peak (9.7 mL, *Figure 1—figure supplement 2A*) were pooled and immediately used for NMR measurements.

## Purification of histones $Sp$H3−$Sp$H4

Cells expressing $Sp$H3, $Sp$H4 with 6His-dAsf1 were pelleted by centrifugation and resuspended in the lysis buffer LB5 (50 mM Tris-HCl pH 8, 500 mM NaCl, 5% glycerol, 1% Triton X-100, 1 mM DTT, 10 mM MgCl2, 0.5 mM PMSF, 1 X cOmplete EDTA-free Protease Inhibitor Cocktail) and flash frozen in liquid nitrogen. After thawing, lysozyme and benzonase were added at a final concentration of 0.25 mg/mL and 70 U/mL respectively. After incubation 20 min at 4 °C, cells were lysed by sonication. Soluble 6His-Asf1 was removed on a NiNTA column (QIAGEN) equilibrated in the LB5 buffer. The flow through (containing soluble-free histones) was filtered with 0.22 µ filters and loaded on a cation exchange Resource S column (GE Healthcare) equilibrated with the dilution buffer EB5_1 (50 mM Tris-HCl pH8). Histones H3−H4 were eluted with a NaCl gradient in a buffer EB5_2 (50 mM Tris-HCl pH8, 2 M NaCl). The H3−H4-containing fractions were pooled, the salt concentration adjusted to 2 M NaCl, and concentrated in a 3 kDa concentrator (Millipore), flash freezed in liquid nitrogen and stored at –70 °C.

## Purification of Pcf1_ED and Pcf1_ED*

Cells expressing Pcf1_ED or Pcf1_ED* with a N-terminal 6His-SUMO tag were collected by centrifugation, resuspended in lysis buffer LB6 (50 mM Tris-HCl pH8, 500 mM NaCl, 5% glycerol, 1% Triton X-100, 1 mM PMSF, 1 µM aprotinin, 0.25 mM DTT) and flash frozen in liquid nitrogen. After thawing, lysosyme was added at a final concentration of 1 mg/mL and cells were incubated 30 min at 4 °C and lysed by sonication. 6His-SUMO-Pcf1_ED was first purified on Histrap colums (Cytiva). Fractions containing the protein were pulled. SUMO protease was added at a final concentration 1/10 and the mixture was dialyzed overnight at 4 °C against the buffer DB6 (50 mM Tris-HCl pH 8, 150 mM NaCl, 10 mM imidazole) and applied on a NiNTA column (QIAGEN) equilibrated in the DB6 buffer.

The flow-through fraction containing Pcf1_ED or Pcf1_ED* was then purified by size exclusion chromatography using a Superdex 75 increase 10/300 column (Cytiva) previsouly equilibrated with the final buffer in FB6 (10 mM Tris-HCl, 50 mM HEPES pH 7, 300 mM NaCl). Fraction containing Pcf1_ED or Pcf1_ED* were concentrated using Amicon centrifuge filter units of 3 kDa cutoff (Millipore) flash freezed in liquid nitrogen and stored at –20 °C or –70 °C.

## Purification of Pcf1_KER and Pcf1_KER-PIP

Cells expressing Pcf1_KER(56-170), or Pcf1-KER-PIP(56-185) (WT or mutant) with a N-terminal 6His-MBP-TEV tag were collected by centrifugation, resuspended in lysis buffer LB7 (50 mM Tris-HCl pH 8, 500 mM NaCl, 5% glycerol, 1% Triton X-100, 1 mM PMSF, 1 µM aprotinin, 0.25 mM DTT, 1 X cOmplete EDTA-free Protease Inhibitor Cocktail) and flash frozen in liquid nitrogen. After thawing, 5 mM MgCl2, 1 mg/mL lysozyme and 70 U/mL of benzonase were added and cells were lysed by sonication. Proteins were first purified on Histrap columns (Cytiva) including a wash step with WB7 (50 mM Tris-HCl pH 8, 1000 mM NaCl). 1 mM DTT and TEV protease (1/10 ratio) was added to the fractions containing the 6His-MBP-TEV_ Pcf1_KER fragment and the mixture was incubated 2 hours at room temperature and injected on a resource S column (Cytiva) previously equilibrated with EB7_1 (50 mM Tris-HCl pH 8). A gradient was applied with the high-salt buffer EB7_2 (50 mM Tris-HCl pH 8, 2 M NaCl). Fractions containing Pcf1_KER fragment were pooled and diluted to reach a concentration of 150 mM NaCl and concentrated (with Amicon Ultra-10 kDa filter concentrators).

## Purification of Pcf1_WHD

Cells expressing Pcf1_WHD with a N-terminal 6His-SUMO tag were resuspended in lysis buffer LB8 (50 mM Tris-HCl pH8, 500 mM NaCl, 5% glycerol, 1% Triton X-100, 1 mM PMSF, 1 µM aprotinin, 0.25 mM DTT, 1 X cOmplete EDTA-free Protease Inhibitor Cocktail) and flash frozen in liquid nitrogen. After thawing, 5 mM MgCl2, 1 mg/mL lysozyme and 70 U/mL of benzonase was added and cells were further lysed by sonication. The lysate was loaded onto gravity flow amylose resin (NEB) previously equilibrated with buffer WB8_1 (50 mM Tris-HCl pH 8). Resin was then washed with 10 column volume of buffer WB8_1, 10 column volumes of buffer WB8_2 (50 mM Tris-HCl pH 8, 1000 mM NaCl), 10 column volumes of buffer WB8_1. 6His-SUMO- Pcf1_WHD was eluted with 10 column volume of buffer EB8 (50 mM Tris-HCl pH 8, 500 mM NaCl, 250 mM Imidazole). SUMO protease was added at a final concentration 1/10 and the mixture was incubated overnight at 4 °C. The mixture was concentrated (with Amicon Ultra-3 kDa filter concentrators) and applied on a Superdex 75 increase 10/300 size exclusion column (Cytiva) previously equilibrated with the final buffer FB8 (10 mM Tris-HCl, 50 mM HEPES pH7, 150 mM, NaCl). Finally, proteins were concentrated in a 3 kDa concentrator (Millipore).

## Purification of SpPCNA

Cells expressing SpPCNA with a N-terminal 6His-SUMO tag were resuspended in lysis buffer LB9 (50 mM Tris-HCl pH8, 500 mM NaCl, 5% glycerol, 1% Triton X-100, 1 mM PMSF, 1 µM aprotinin, 0.25 mM DTT, 1 X cOmplete EDTA-free Protease Inhibitor Cocktail) and flash frozen in liquid nitrogen. After thawing, 1 mM MgCl2, 1 mg/mL lysozyme and 70 U/mL of benzonase was added and cells were further lysed by sonication. The lysate was loaded onto gravity flow amylose resin (NEB) previously equilibrated with buffer WB9_1 (50 mM Tris-HCl pH 8). Resin was then washed with 10 column volume of buffer WB9_1, 10 column volumes of buffer WB8_2 (50 mM Tris-HCl pH 8, 2000 mM NaCl), 10 column volumes of buffer WB8_1. 6His-SUMO-SpPCNA was eluted with 3 column volumes of buffer EB9 (50 mM Tris-HCl pH 8, 250 mM Imidazole). One mM DTT and SUMO protease was added at a final concentration 1/10 and the mixture was dialyzed overnight at 4 °C against the buffer DB9 (50 mM Tris-HCl pH 8, 150 mM NaCl, 10 mM imidazole). The mixture was applied on a Histrap column (Cytiva), the flow through (containing SpPCNA) was concentrated (with Amicon Ultra-15 3 kDa filter concentrators) and applied on a HiLoad 16/600 superdex 200 size exclusion column previously equilibrated with a FB9 (50 mM Tris-HCl pH8, 150 mM NaCl). In case the digestion of the tag was incomplete, the two last steps digestion with SUMO protease and gel filtration were repeated.

For all protein samples, depending on specific requirements of different techniques used, aliquots of concentrated protein were either maintained at 4 °C or flash frozen in liquid nitrogen after addition or not of 30% glycerol and stored at –70 °C for further use.

## DNAs used to monitor protein-DNA interactions

The different DNAs were purchased from eurofins genomics. The sequences were derived from the 601 positioning sequence: ATCAATATCCACCTGCAGATACTACCAAAAGTGTATTTGG. For MST, the DNA were labeled with ALEXA488 at their 5' extremity. The ssDNA was annealed with the reverse-complementary sequence by heating at 90 °C and cooling slowly at room temperature.

## Size-exclusion chromatography (SEC)

*Sp*CAF-1 subunits interaction was performed by mixing 2.2 nmoles of each isolated protein together in a final volume of 1.26 mL and left overnight at 5 °C. The complexes were then concentrated to 300 µL (with Amicon Ultra-15 30 kDa filter concentrators) and injected into a Superdex 200 increase 10/300 (Cytiva) for separation by size-exclusion chromatography previously equilibrated with the FB4_3 (50 mM Tris-HCl pH 7.5, 500 mM NaCl, 1 mM DTT). The different fractions were analyzed on mPAGE 8% Bis-Tris Precast Gels (Sigma) with MOPS SDS running buffer. Interaction between CAF-1 and H3−H4 was carried out by incubating for 3 hr, 3 nmoles of *Sp*CAF-1 with 3 nmoles of *Sp*H3−H4 in a final buffer FB4_4 (50 mM Tris-HCl pH 7.5, 150 mM NaCl, 4 mM DTT, 1 X cOmplete EDTA-free Protease Inhibitor Cocktail) or FB4_5 (50 mM Tris-HCl pH 7.5, 1 M NaCl, 4 mM DTT, 1 X cOmplete EDTA-free Protease Inhibitor Cocktail). Samples were then concentrated to 300 µL (with Amicon Ultra-15 30 kDa filter concentrators) and injected into a Superdex 200 increase 10/300 (Cytiva) for separation by size-exclusion chromatography with their corresponding buffers. The different fractions were analyzed on mPAGE 4–20% Bis-Tris Precast Gels (Sigma) with MES SDS running buffer.

## Electrophoretic Mobility Shift Assay (EMSA)

The proteins and DNA were mixed to be in a final EMSA buffer EMB (25 mM Tris-HCl pH 8, 1 mM EDTA pH 8.0, 150 mM NaCl) and incubated at 4 °C for 30 min and heated at 37 °C for 5 min prior to analysis on precast "any KD" Mini-PROTEAN TGX (Bio-Rad, Cat #4569033) polyacrylamide gels using 1 x TBE as running buffer. The gels were stained with 1 x of SYBR Safe (Thermo Fisher Scientific, Waltham, MA) then visualized with BIORAD EZ Imager. A second identical the gel was Coomassie Blue before being visualized with the BIORAD EZ Imager. Band intensities were quantified by ImageJ.

## MicroScale thermophoresis (MST)

DNAs labeled with ALEXA488 at their 5' extremity were adjusted to 20 nM in the final dilution buffer FB_MST (10 mM Tris-HCl, 40 mM HEPES pH 7, 150 mM NaCl). Freshly prepared proteins or complexes were diluted in the same buffer with 16 serial 1:2 dilutions. Each protein dilution was mixed with one volume of labeled DNA and filled into Monolith NT standard treated capillaries (NanoTemper Technologies GmbH). Thermophoresis was measured using a Monolith NT.115 instrument (NanoTemper Technologies GmbH) at an ambient temperature of 20 °C with 3 s/20 s/1 s laser off/on/off times, respectively. Instrument parameters were adjusted with 80% LED power and 40% MST power. Data of two measurements were analyzed (MO.Affinity Analysis software, NanoTemper Technologies) using the signal from thermophoresis at 5 s. The fits were performed with a Hill model calculating effective concentration at which a 50% signal is seen (EC50) and a Hill coefficient giving an estimation of the cooperativity of the reaction (*Tso et al., 2018*).

## Circular dichroism (CD)

Circular dichroism (CD) measurements were carried out at 20 °C on a JASCO J-810 spectro-polarimeter. Temperature was controlled by a Peltier. Spectra from 190 to 250 nm were obtained using a 2 mm optical path length quartz cell (Hellma #100-2-40) containing Pcf1_KER or Pcf1_KER* (5 µM) in 10 mM of phosphate buffer (pH 7.4).

## Nuclear magnetic resonance (NMR)

NMR experiments were carried out on Bruker DRX-600 MHz, 700 MHz or 950MHz spectrometers equipped with cryo-probes. All NMR data were processed using Topspin (Bruker) and analyzed using Sparky (T.D. Goddard and D.G. Kneller, UCSF). Samples were prepared in 3 mm NMR tubes, in solution containing 5% D2O, 0.1% NaN3, 0.1 mM DSS with different buffer appropriate for different complex formations or reactions. Heteronuclear Multiple Quantum Correlation (sofast-HMQC) or best-HSQC spectra were all recorded at 283°K. The protein concentrations were between 9 µM and 500 µM.

For backbone resonances assignments, 3D data were collected at 283°K using standard Heteronuclear Single Quantum Correlation (HSQC) spectra $^1$H-$^{15}$N HSQC, TOCSY-HSQC, HNCA, HBHA(CO)NH, CBCA(CO)NH, HN(CA)CO, HNCO, HN(CO)CA, CBCANH and HN(CA)CO experiments. Proton chemical shifts (in ppm) were referenced relative to internal DSS and $^{15}$N and $^{13}$C references were set indirectly relative to DSS using frequency ratios (**Wishart et al., 1995a**). Chemical shift index were calculated according to the sequence-specific random coil chemical shifts (**Wishart et al., 1995b**, **Tamiola et al., 2010**).

Structural models of the SpCAF-1 WHD domain were computed from NMR data with CS-ROSETTA (**Shen et al., 2008**) version 1.01. First, the MFR program from NMRpipe (**Delaglio et al., 1995**) was used to search a structural database for best matched fragments based on the protein backbone $^{15}$N, $^{13}$C, $^{13}$CA, $^{13}$CB, and $^1$HN chemical shifts. Then the ROSETTA 3.8 software was used to generate 27,753 models by fragment assembly and full-atom relaxation. These models were rescored by comparing the experimental chemical shifts with the chemical shifts predicted by SPARTA (**Shen and Bax, 2007**) for each model. The best model after rescoring was chosen as a representative NMR model of the WHD domain.

## Small angle X-ray scattering (SAXS)

SAXS data were collected at the SWING beamline on a EigerX 4 M detector using the standard beamline setup in SEC mode (**Thureau et al., 2021**). Samples were injected into a Superdex 5/150 GL (Cytivia) column coupled to a high-performance liquid chromatography system, in front of the SAXS data collection capillary. The initial data processing steps including masking and azimuthal averaging were performed using the program FOXTROT (**Evans et al., 2022**) and completed using US-SOMO (**Brookes et al., 2016**). The final buffer subtracted and averaged SAXS profiles were analyzed using ATSAS v.3. software package (**Manalastas-Cantos et al., 2021**). To model the structures and improve the AlphaFold2 models, the program Dadimodo (**Rudenko et al., 2019**) (https://dadimodo.synchrotron-soleil.fr) that refines multidomain protein structures against experimental SAXS data was used (see **Supplementary file 1a** for more information).

## Structural modeling

Sequences of *S. pombe* Pcf1 (Q1MTN9), Pcf2 (O13985), Pcf3 (Q9Y825), H3 (P09988), H4 (P09322), and PCNA (Q03392) were retrieved from UniProt database **UniProt, 2021**. These sequences were used as input of mmseqs2 homology search program **Steinegger and Söding, 2017** used with three iterations to generate a multiple sequence alignment (MSA) against the uniref30_2103 database (**Mirdita et al., 2022**). The resulting alignments were filtered using hhfilter (**Steinegger et al., 2019**) using parameters ('id'=100, 'qid'=25, 'cov'=50) and the taxonomy assigned to every sequence keeping only one sequence per species. To increase the number of sequences in the alignment of *S. pombe* Pcf1, we independently generated MSA using mmseqs2 starting from the *S. cerevisiae* or the human homolog of Pcf1 (Q12495 and Q13111, respectively) and the resulting alignments were combined with the one of SpPcf1. Full-length sequences in the alignments were then retrieved and the sequences were realigned using MAFFT (**Katoh and Standley, 2013**) with the default FFT-NS-2 protocol. To build the so-called mixed co-alignments, sequences in the alignment of individual partners were paired according to their assigned species and left unpaired in case no common species were found (**Mirdita et al., 2022**). A first global model with full-length Pcf1, Pcf2 and Pcf3 was generated to map the regions of Pcf1 binding to Pcf2 and Pcf3 and to obtain the pLDDT scores shown in **Figure 1B** for Pcf1, **Figure 1—figure supplement 2E** for Pcf2 and Pcf3. Next, three models of the complex corresponding to independent modules of the complex were generated using different delimitations: model_1 (presented in **Figure 1—figure supplement 3**) with Pcf1(403-450)-Pcf2(1-453) (MSA with 2180 species, 501 positions), model_2 (presented in **Figure 1—figure supplement 4**) with Pcf1(200-335)-Pcf3(1-408) (MSA with 2148 species, 544 positions), model_3 Pcf1(352-383)-H3(60–136)-H4(25–103) (presented in **Figure 2F** and **Figure 2—figure supplement 2A**) (MSA with 3530 species, 188 positions). Concatenated mixed MSAs were generated using the delimitations defined above and used as input to run 5 independent runs of the Alphafold2 algorithm with 6 iterations each (**Jumper et al., 2021**) generating 5 structural models using a local version of the ColabFold interface (**Mirdita et al., 2022**) trained on the multimer dataset (**Evans et al., 2022**) on a local HPC equipped with NVIDIA Ampere A100 80Go GPU cards. The best models of each of the 5 runs converged toward

similar conformations. They reached high confidence and quality scores with pLDDTs in the range [83.7, 84.3], [88.8, 89.8], and [86.5, 88.4] and the model confidence score (weighted combination of pTM- and ipTM-scores with a 20:80 ratio) *Evans et al., 2022* in the range [0.9, 0.93], [0.88, 0.89], [0.85, 0.87], for model_1, model_2, and model_3, respectively. The models with highest confidence score for each of the three models were relaxed using rosetta relax protocols to remove steric clashes (*Leman et al., 2020*) with constraints (std dev. of 2 Å for the interatomic distances) and were used for structural analysis. MSA web logos were generated with the weblogo server (https://weblogo.berkeley.edu/logo.cgi).

## Nucleosome assembly assay

Mock- and p150CAF-1-depleted *Xenopus* high-speed egg extract (HSE) were prepared as previoulsy (*Ray-Gallet and Almouzni, 2004*). Nucleosome assembly was performed on pBS plasmid damaged by UV (500 J/m2) to promote DNA synthesis as previously described (*Ray-Gallet and Almouzni, 2004*) except that the reaction mixed contained 3.2 µM of biotin-14-dCTP (Invitrogen, Ref 19518–0189) instead of [$\alpha^{32}$P]-dCTP. The p150CAF-1-depleted extracts were complemented with 50 ng of isolated/reconstituted *Sp*CAF-1 complex composed of WT or mutated Pcf1. After DNA purification, samples were by processed for gel electrophoresis (1% agarose) to resolve topoisomers as previously described (*Ray-Gallet and Almouzni, 2004*). After staining with Ethidium bromide to visualize total DNA and gel transfer on a Nylon N+membrane (GE Healthcare Ref RPN203B) (Qbiogen) for 45 min at 40 mbar in 10 x SSC, the membrane was rinsed in PBS, air dried and DNA was crosslinked to the membrane using Stratalinker (Bio-Rad). DNA synthesis was visualized by detecting biotin with the Phototope-Star detection kit (New England Biolabs Ref N7020S) and images acquired on a Chemidoc system (Bio-Rad).

## Standard yeast genetics

Yeast strains were freshly thawed from frozen stocks and grown at 30 °C using standard yeast genetics practices. All *pcf1* mutants were obtained using a two-step transformation approach in which the region of interest of the coding sequence was first replaced by the ura4+ marker and then by synthetic DNA containing the appropriated mutations. Therefore, all mutations are marker-free and were followed in genetic analysis by PCR and sequencing. Yeast strains used in this study are listed in *Supplementary file 1b*. For tetrad analysis, the genotypes of viable spores were deduced by checking for resistance to Kanamycin (deletion) and sequencing, which made it possible to deduce the genotype of dead spores for each tetrad. At least 18 tetrads were analyzed for each cross.

## Peri-centromeric silencing assay

5-FOA (EUROMEDEX, 1555) resistant colonies were grown on uracil-containing liquid media overnight and 10 µL of fivefold serial dilutions (from $1.10^7$ cells/mL to $1.10^5$ cells/ml) were spotted on indicated media.

## Co-immunoprecipitation

A total of $5.10^8$ cells from exponentially growing cultures were harvested with 10% NaN$_3$ and 1 mM PMSF, final concentration, and then washed twice in water and once in Lysis buffer (buffer 50 mM HEPES High salt, 50 mM KoAc pH7.5, 5 mM EGTA, 1% triton X100, 0.01 mg/mL AEBSF, EDTA-free protease inhibitor cocktail). Cell pellets were resuspended in 800 µL of lysis buffer and were broken with a Precellys homogenizer (twice 4 cycles at 10 000 rpm, 20 sec-2 min pause). After lysate clarification (30 min at 13000 rpm, 4 °C), 2.5 mg of proteins were incubated with pre-washed Dynabeads protein G (Invitrogen, 10003D) coupled to anti-FLAG antibody (Sigma F7425) and incubated overnight at 4 °C on a wheel. Beads were washed three times for 5 min at 4 °C with 800 µL of lysis buffer, and then resuspended in 1 X Laemmli buffer, and boiled at 95 °C for 10 min. INPUT and UNBOUND (both 10% of initial protein extract) and BOUND (IP) fraction were resolved by electrophoresis on acrylamide gels (4–12% Invitrogen) and then transferred onto nitrocellulose membrane that were saturated for 1 hr, RT in TBS-0.075% tween-5% milk. Proteins of interest were detected with anti-FLAG antibody (Sigma F1805, 1:1000) and anti-PCNA antibody (Santa Cruz sc-8349, 1:500).

## Live cell imaging

All image acquisition was performed on the PICT-IBiSA Orsay Imaging facility of Institut Curie. Cells were grown in filtered supplemented EMM-glutamate. Exponentially growing cultures were

centrifuged and resuspended in 50 µL of fresh medium. Two µL from this concentrated solution was dropped onto a Thermo Scientific slide (ER-201B-CE24) covered with a thin layer of 1.4% agarose in filtered EMMg. 13 z-stack pictures (each z step of 300 nm) were captured using a Spinning Disk Nikon inverted microscope equipped with the Perfect Focus System, Yokogawa CSUX1 confocal unit, Photometrics Evolve512 EM-CCD camera, 100 X/1.45-NA PlanApo oil immersion objective and a laser bench (Errol) with 491 (GFP) and 561 (MmCherry) nm diode lasers, 100 mX (Cobolt). Pictures were collected with METAMORPH software and analyzed with ImageJ. For Pcf2-GFP and Rad52-GFP foci, a threshold (using the find maxima tool, >400 for Pcf2-GFP foci and >100 for Rad52-GFP) was setup at the same level for each genetic background analyzed within the same experiment. Images from *Figure 6C* were deconvolved using the Huygens remote manager software.

## Acknowledgements

We thank the Alain LECOQ and Denis SERVENT from giving access to the CD spectro-polarimeter. This work was supported by grants from the INCA (2016–1-PL BIO-03-CEA-1, 2016–1-PLBIO-03-ICR-1), ANR (ANR-16-CE11-0028; ANR-20-CE18-0038; ANR-21-CE11-0027; ANR-21-CE35-0013) the program labeled by the ARC foundation 2016 (PGA1*20160203953), the Fondation LIGUE "Equipe Labellisée 2020" (EL2020LNCC/Sal), and by french infrastructures, the Synchrotron SOLEIL (20191119; 20210745), the French Infrastructure for Integrated Structural Biology (FRISBI) ANR-10-INBS-0005 and the IR INFRANALYTICS FR2054. It benefited from the ERC-2015-ADG-694694 "ChromADICT", the Ligue Nationale contre le Cancer (Equipe labellisée Ligue) and ANR-11-LABX-0044 5. We also thank the PICT-IBiSA@Orsay Imaging Facility of the Institut Curie (particularly Laetitia Besse).

## Additional information

### Funding

| Funder | Grant reference number | Author |
| --- | --- | --- |
| Institut National Du Cancer | 2016-1-PL BIO-03-CEA1 | Maxime Audin |
| Agence Nationale de la Recherche | ANR-16-CE11-0028 | Fouad Ouasti |
| Agence Nationale de la Recherche | ANR-20-CE18-0038 | Mehdi Tachekort |
| Institut National Du Cancer | 2016-1-PLBIO-03-ICR-1 | Ibrahim Soumana Adamou |

The funders had no role in study design, data collection and interpretation, or the decision to submit the work for publication.

### Author contributions

Fouad Ouasti, Maxime Audin, Investigation, Writing – original draft, Data curation, Formal analysis, Visualization; Karine Fréon, Elizabeth Cesard, Gwenaelle Moal, Ibrahim Soumana Adamou, Aleksandra Uryga, Investigation; Jean-Pierre Quivy, Conceptualization, Funding acquisition, Investigation, Writing – original draft, Writing – review and editing, Visualization; Mehdi Tachekort, Investigation, Visualization; Aurélien Thureau, Methodology, Formal analysis; Virginie Ropars, Paloma Fernández Varela, Pierre Legrand, Methodology; Jessica Andreani, Raphaël Guerois, Investigation, Methodology; Geneviève Almouzni, Funding acquisition, Writing – review and editing, Conceptualization; Sarah Lambert, Conceptualization, Supervision, Writing – review and editing, Data curation, Formal analysis, Funding acquisition, Project administration, Visualization, Writing – original draft; Francoise Ochsenbein, Conceptualization, Supervision, Investigation, Writing – original draft, Project administration, Writing – review and editing, Data curation, Formal analysis, Funding acquisition, Visualization

### Author ORCIDs

Karine Fréon http://orcid.org/0000-0001-7853-078X
Jean-Pierre Quivy https://orcid.org/0000-0001-6557-7204
Jessica Andreani http://orcid.org/0000-0003-4435-9093
Raphaël Guerois http://orcid.org/0000-0001-5294-2858

Sarah Lambert https://orcid.org/0000-0002-1403-3204
Francoise Ochsenbein https://orcid.org/0000-0002-9027-4384

Reviewer #1 (Public Review): https://doi.org/10.7554/eLife.91461.3.sa1
Reviewer #2 (Public Review): https://doi.org/10.7554/eLife.91461.3.sa2
Reviewer #3 (Public Review): https://doi.org/10.7554/eLife.91461.3.sa3
Author response https://doi.org/10.7554/eLife.91461.3.sa4

## Additional files

### Supplementary files
MDAR checklist

Supplementary file 1. Supplementary tables. (A) Experimental information and modelling of SAXS data. (B) Yeast strains used in this study.

### Data availability
We deposited structural models generated by AlphaFold2 at the modelarchive repository site (https://doi.org/10.5452/ma-1bb5w, https://doi.org/10.5452/ma-bxxkp, https://doi.org/10.5452/ma-htx0n).

The following datasets were generated:

| Author(s) | Year | Dataset title | Dataset URL | Database and Identifier |
|---|---|---|---|---|
| Guerois R, Ochsenbein F | 2023 | Model of the complex between the histone chaperones Pcf1 (ortholog of CAF1 subunit A in human) and Pcf2 (ortholog of CAF1 subunit B in human) in *S. pombe* | https://doi.org/10.5452/ma-1bb5w | ModelArchive, 10.5452/ma-1bb5w |
| Guerois R, Ochsenbein F | 2023 | Model of the complex between the histone chaperones Pcf1 (ortholog of CAF1 subunit A in human) and Pcf3 (ortholog of CAF1 subunit C in human) in *S. pombe* | https://doi.org/10.5452/ma-bxxkp | ModelArchive, 10.5452/ma-bxxkp |
| Guerois R, Ochsenbein F | 2023 | Model of the complex between the histone chaperones Pcf1 (ortholog of CAF1 subunit A in human) and histones H3 and H4 in *S. pombe* | https://doi.org/10.5452/ma-htx0n | ModelArchive, 10.5452/ma-htx0n |

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
