## [Editor Report · eLife assessment]

This **important** study advances our understanding of the machinery that couples DNA synthesis with the deposition of histone proteins onto newly synthesized DNA. A **convincing** array of experiments combines NMR, protein biochemistry, and in vivo analyses of Chromatin Assembly Factor-1 of fission yeast. The work is of interest to researchers in the field of chromosome/chromatin biology as well as epigenetics.

---

## [Referee Report · Reviewer #1 (Public Review)]

Summary:

This paper makes important contributions to the structural analysis of the DNA replication-linked nucleosome assembly machine termed Chromatin Assembly Factor-1 (CAF-1). The authors focus on the interplay of domains that bind DNA, histones and replication clamp protein PCNA.

Strengths:

The authors analyze soluble complexes containing full-length versions of all three fission yeast CAF-1 subunits, an important accomplishment given that many previous structural and biophysical studies have focused on truncated complexes. New data here supports previous experiments indicating that the KER domain is a long alpha helix that binds DNA. Via NMR, the authors discover structural changes at the histone binding site, defined here with high resolution. Most strikingly, the experiments here show that for the *S. pombe* CAF-1 complex, that the WHD domain at the C-terminus of the large subunit lacks DNA binding activity observed in the human and budding yeast homologs, indicating a surprising divergence in the evolution of this complex. Together, these are important contributions to the understanding of how the CAF-1 complex works.

Weaknesses:

1. Given the strong structural predication about the roles of residues L359 and F380 (Fig. 2f), mutation of these residues would be the definitive test of their contribution to histone binding.

2. Could it be that the apparent lack of histone deposition by the delta-WHD mutant complex occurs because this mutant complex is unstable when added to the *Xenopus* extract?

---

## [Referee Report · Reviewer #2 (Public Review)]

Summary:

The authors describe the structure-functional relationship of domains in *S. pombe* CAF-1, which promotes DNA replication-coupled deposition of histone H3-H4 dimer. The authors nicely showed that the ED domain with an intrinsically disordered structure binds to histone H3-H4, that the KER domain binds to DNA and that, in addition to a PIP box, the KER domain also contributes to the PCNA binding. The ED and KER domains as well as the WHD domain are essential for nucleosome assembly in vitro. The ED, KER domains and the PIP box are important for the maintenance of heterochromatin.

Strengths:

The combination of structural analysis using NMR and Alphafold2 modeling with biophysical and biochemical analysis provided strong evidence on the role of the different domain structures of the large subunit of SpCAF-1, spPCF-1 in the binding to histone H3-H4, DNA as well as PCNA. The conclusion was further supported by genetic analysis of the various pcf1 mutants. The large amounts of data provided in the paper support the authors' conclusion very well.

Weaknesses:

---

## [Referee Report · Reviewer #3 (Public Review)]

Summary: The study conducted by Ouasti et al. is an elegant investigation of fission yeast CAF-1, employing a diverse array of technologies and genetic alterations to dissect its functions and their interdependence. These functions play a critical role in specifying interactions vital for DNA replication, heterochromatin maintenance, and DNA damage repair, and their dynamics involve multiple interactions. The authors have extensively utilized various in vitro and in vivo tools to validate their model and emphasize the dynamic nature of this complex.

Strengths: Their work is supported by robust experimental data from multiple techniques, including NMR and SAXS, which validate their molecular model. They conducted in vitro interactions using EMSA and isothermal microcalorimetry, in vitro histone deposition using *Xenopus* high-speed egg extract, and systematically generated and tested various genetic mutants for functionality in in vivo assays. They successfully delineated domain-specific functions using in vitro assays and could validate their roles to large extent using genetic mutants. One significant revelation from this study is the unfolded nature of the acidic domain, observed to fold when binding to histones. Additionally, the authors also elucidated the role of the long KER helix in mediating DNA binding and enhancing the association of CAF-1 with PCNA. The paper effectively addresses its primary objective.

Weaknesses: A few relatively minor unresolved aspects persist, which, if clarified or experimentally addressed by the authors, could further bolster the study.

1. The precise function of the WHD domain remains elusive. Its deletion does not result in DNA damage accumulation or defects in heterochromatin maintenance. This raises questions about the biological significance of this domain and whether it is dispensable. While in vitro assays revealed defects in chromatin assembly using this mutant (Figure 5), confirming these phenotypes through in vivo assays would provide additional assurance that the lack of function is not simply due to the in vitro system lacking PTMs or other regulatory factors.

2. The observation of increased Pcf2-gfp foci in pcf1-ED* cells, particularly in mono-nucleated (G2-phase) and bi-nucleated cells with septum marks (S-phase), might suggest the presence of replication stress. This could imply incomplete replication in specific regions, leading to the persistence of Caf1-ED*-PCNA factories throughout the cell cycle. To further confirm this, detecting accumulated single-stranded DNA (ssDNA) regions outside of S-phase using RPA as an ssDNA marker could be informative.

3. Moreover, considering the authors' strong assertion of histone binding defects in ED* through in vitro assays (Figure 2d and S2a), these claims could be further substantiated, especially considering that some degree of histone deposition might still persist in vivo in the ED* mutant (Figure 7d, viable though growth defective double ED*+hip1D mutants). For example, the approach, akin to the one employed in Fig. 6a (FLAG-IPs of various Pcf1-FLAG-tagged mutants), could also enable a comparison of the association of different mutants with histones and PCNA, providing a more thorough validation of their findings.

4. It would be valuable for the authors to speculate on the necessity of having disordered regions in CAF1. Specifically, exploring the overall distribution of these domains within disordered/unfolded structures could provide insightful perspectives. Additionally, it's intriguing to note that the significant disparities observed among mutants (ED*, PIP*, and KER*) in in vitro assays seem to become more generic in vivo, except for the indispensability of the WHD-domain. Could these disordered regions potentially play a crucial role in the phase separation of replication factories? Considering these questions could offer valuable insights into the underlying mechanisms at play.

---

## [Author Response]

The following is the authors’ response to the original reviews.

We thank the three reviewers and the reviewing editor for their positive evaluation of our manuscript. We particularly appreciate that they unanimously consider our work as “important contributions to the understanding of how the CAF-1 complex works”, “The large amounts of data provided in the paper support the authors' conclusion very well” and “The paper effectively addresses its primary objective and is strong”. We also thank them for a careful reading and useful comments to improve the manuscript. We have built on these comments to provide an improved version of the manuscript, and address them point by point below.

**Reviewer #1 (Public Review):**
Summary:This paper makes important contributions to the structural analysis of the DNA replication-linked nucleosome assembly machine termed Chromatin Assembly Factor-1 (CAF-1). The authors focus on the interplay of domains that bind DNA, histones, and replication clamp protein PCNA.Strengths:The authors analyze soluble complexes containing full-length versions of all three fission yeast CAF-1 subunits, an important accomplishment given that many previous structural and biophysical studies have focused on truncated complexes. New data here supports previous experiments indicating that the KER domain is a long alpha helix that binds DNA. Via NMR, the authors discover structural changes at the histone binding site, defined here with high resolution. Most strikingly, the experiments here show that for the *S. pombe* CAF-1 complex, the WHD domain at the C-terminus of the large subunit lacks DNA binding activity observed in the human and budding yeast homologs, indicating a surprising divergence in the evolution of this complex. Together, these are important contributions to the understanding of how the CAF-1 complex works.Weaknesses:1. There are some aspects of the experimentation that are incompletely described:

We thank the reviewer for his/her careful reading of the manuscript. Indeed, we plotted two curves in Figure S1C in a color that does not match the legend, leading to confusion. Curve 1, Pcf1 alone, depicted in red, should appear in pink as indicated in the legend and in the SDS-PAGE analysis below. Curve 1 exhibits two peaks, labeled as 1a and 1b. With an elution volume of 8.5mL close to the dead volume of the column, peak 1a corresponds to soluble oligomers, while peak 1b (10.4mL) likely corresponds to monomeric Pcf1. Curve 5 (Pcf1 + Pcf2 mixture) was in pink instead of purple as indicated in the legend. This curve consists of three distinct peaks (5a, 5b, and 5c). The SDS-PAGE analysis revealed the presence of oligomers of Pcf1-Pcf2 (5a, 8.3mL), the Pcf1-Pcf2 complex (5b, 9.8mL), and Pcf2 alone (5c, 13.6 mL).

The color has now been corrected in the revised manuscript.

More importantly, was a particular SEC peak of the three-subunit CAF-1 complex (i.e. 4a or 4b) characterized in the further experimentation, or were the data obtained from the input material prior to the separation of the different peaks? If the latter, how might this have affected the results? Do the forms inter-convert spontaneously?

We conducted all structural analyses and DNA/PCNA interactions Figures (1-4, S1-S4) with freshly SECpurified samples corresponding to the 4b peak (9.7mL). Aliquots were flash-frozen with 50% glycerol for in vitro histone assembly assays (Figure 5).

1. Given the strong structural predication about the roles of residues L359 and F380 (Fig. 2f), these should be mutated to determine effects on histone binding.

We are pleased that our structural predictions are considered as strong. We agree that investigating the role of the L359 and F380 residues will be critical to further refine the binding interface between histone H3-H4 and CAF-1. An in vitro and in vivo analysis of such mutated forms, alongside the current Pcf1-ED mutant characterized in this article and additional potential mutated forms, has the potential to provide a better understanding of the dynamic of histone deposition by CAF-1. However, these additional approaches would require to reach another step in breaking this enigmatic dynamic.

1. Could it be that the apparent lack of histone deposition by the delta-WHD mutant complex occurs because this mutant complex is unstable when added to the *Xenopus* extract?

We cannot formally exclude this possibility, and this could potentially applies to all mutated forms tested. However, in the absence of available antibodies against the fission yeast CAF-1 complex, we cannot test this hypothesis for technical reasons. Nevertheless, we feel reassured by the fact that the in vitro assays of nucleosome assembly are overall consistent with the in vivo assays. Indeed, all mutated forms tested that abolished or weakened nucleosome assembly also exhibited synthetic lethality/growth defect in the absence of a functional HIRA pathway, including the delta WHD mutated form. This genetic synergy, that reflects a defective histone deposition by CAF-1, is not specific to the fission yeast *S. pombe* and was previously reported in S. cerevisiae (Kaufman et al. MCB 1998; Krawitz et al. MCB 2002). This further supports the evolutionary conservation based on genetic assay as a read out for defective histone deposition by CAF-1.

**Reviewer #1 (Recommendations For The Authors):**
p. 4: "An experimental molecular weight of 179 kDa was calculated using Small Angle X-ray Scattering (SAXS), consistent with a 1:1:1 stoichiometry (Figure S1e). These data are in agreement with a globular complex with a significant flexibility (Figure S1f)." There needs to be more description of the precision of the molecular weight measurement, and what aspects of these data indicate the flexibility.

The molecular weight was estimated using the correlation volume (Vc) defined by (Rambo & Tainer, Nature 2013, 496, 477-481). The estimated error with this method is around 10%. We added this information together with supporting arguments for the existence of flexibility: “An experimental molecular weight of 179 kDa was calculated using Small Angle X-ray Scattering (SAXS). Assuming an accuracy of around 10% with this method (Rambo and Tainer 2013), this value is consistent with a 1:1:1 stoichiometry for the CAF-1 complex (calculated MW 167kDa) (Figure S1e). In addition, the position of the maximum for the dimensionless Kratky plot was slightly shifted to higher values in the y and x axis compared to the position of the expected maximum of the curve for a fully globular protein (Figure S1f).

This shows that the complex was globular with a significant flexibility.”

p. 6, lines 21-22: "In contrast, a large part of signals (338-396) did not vanish anymore upon addition of a histone complex preformed with two other histone chaperones known to compete with CAF-1 for histone binding..." Given the contrast made later with the 338-351 region which is insensitive to Asf1/Mcm2, it would be clearer for the reader to describe the Asf1/Mcm2-competed regions as residues 325-338 plus 352-396. Note that the numerical scale of residues doesn't line up perfectly with the data points in Figure 2d, and this should be fixed as well.

We thank this reviewer for spotting this typographical error; we intended to write "In contrast, a large part of signals (348-396) did not vanish anymore… “. We modified paragraph as suggested by the reviewer because we agree it is clearer for the reader : “In contrast, only a shorter fragment (338-347) vanished upon addition of Asf1-H3-H4-Mcm2(69-138), a histone complex preformed with two other histone chaperones, Asf1 and Mcm2, known to compete with CAF-1 for histone binding (Sauer et al. 2017) and whose histone binding modes are well established (Figure 2e) (Huang et al. 2015, Richet et al. 2015). This finding underscores a direct competition between residues (325-338) and (349-396) within the ED domain and Asf1/Mcm2 for histone binding.”

The slight shift in the numerical scale Figure 2d was also corrected.

p. 8. Lines 22-24: "EMSAs with a double-stranded 40bp DNA fragment confirmed the homogeneity of the bound complex. When increasing the SpCAF-1 concentration, additional mobility shifts suggest, a cooperative DNA binding (Figure 3a)." I agree that the migration of the population is further retarded upon the addition of more protein. However, doesn't this negate the first sentence? That is, if multiple CAF-1 complexes can bind each dsDNA molecule, can these complexes be described as homogeneous?

We fully agree with the reviewer's comment and have removed the notion of homogeneity from the first sentence. “EMSAs with a double-stranded 40bp DNA fragment showed the formation of a bound complex.”

Figure S2b Legend: "1H-15N HSQC spectra of Pcf1_ED (425-496)." The residue numbers should read 325-396.

The typo has been corrected.

Is the title for Figure 5 correct?: "Figure 5: Rescue using Y340 and W348 in the ED domain, the intact KER DNA binding domain and the C-terminal WHD of Pcf1 in SpCAF-1 mediated nucleosome assembly." I don't see that any point mutation rescue experiments are done here.

The title of figure 5 has been modified for “Efficient nucleosome assembly by SpCAF-1 in vitro requires interactions with H3-H4, DNA and PCNA, and the C-terminal WHD domain”.

Figure S6C. I assume the top strain lacks the Pcf2-GFP but this should be stated explicitly.

The following sentence “The top strain corresponds to a strain expressing wild-type and untagged Pcf2 as a negative control of GFP fluorescence” is now added to the figure legend. The figure S6C has been modified accordingly to mention “Pcf2 (untagged)” and state more explicitly.

Regarding point #3 in the public review, a simple initial test of this idea would be to determine if similar amounts of wt and mutant complexes can be immunoprecipitated at the endpoint of the assembly reactions.

In the absence of available antibodies against the fission yeast CAF-1 complex, we cannot test this hypothesis for technical reasons. However, the in vitro assays of nucleosome assembly are overall consistent with the in vivo assays. Indeed, all mutated forms tested that abolished or weakened nucleosome assembly also exhibited synthetic lethality/growth defect in the absence of a functional HIRA pathway, including the delta WHD mutated form. This genetic synergy, reflecting defective histone deposition by CAF-1, is not specific to the fission yeast *S. pombe*, as it was previously reported in S. cerevisiae (Kaufman et al. MCB 1998; Krawitz et al. MCB 2002), further supporting the evolution conservation in the genetic assay as a read out for defective histone deposition by CAF-1.

Foundational findings that should be cited: The role of PCNA in CAF-1 activity was first recognized by pioneering studies in the Stillman laboratory (PMID: 10052459, 11089978). The earliest recombinant studies of CAF-1 showed that the large subunit is the binding platform for the other two, showed that the KER and ED domains were required for histone deposition activity, and roughly mapped the p60-binding site on the large subunit (PMID: 7600578). Another early study roughly mapped the binding site for the third subunit and showed that biological effects of impairing the PCNA binding synergized with defects in the HIR pathway (PMID: 11756556), a genetic synergy first demonstrated in budding yeast (PMID: 9671489).

We thank the reviewer for providing these important references that are now cited in the manuscript. PMID: 10052459 and 11089978 are cited page 2 line 18 and 19, PMID: 7600578 page 19 line 5 and PMID: 11756556 and 9671489 page 18 line 2.

**Reviewer #2 (Public Review):**
Summary:The authors describe the structure-functional relationship of domains in *S. pombe* CAF-1, which promotes DNA replication-coupled deposition of histone H3-H4 dimer. The authors nicely showed that the ED domain with an intrinsically disordered structure binds to histone H3-H4, that the KER domain binds to DNA, and that, in addition to a PIP box, the KER domain also contributes to the PCNA binding. The ED and KER domains as well as the WHD domain are essential for nucleosome assembly in vitro. The ED, KER domains, and the PIP box are important for the maintenance of heterochromatin.Strengths:The combination of structural analysis using NMR and Alphafold2 modeling with biophysical and biochemical analysis provided strong evidence on the role of the different domain structures of the large subunit of SpCAF-1, spPCF-1 in the binding to histone H3-H4, DNA as well as PCNA. The conclusion was further supported by genetic analysis of the various pcf1 mutants. The large amounts of data provided in the paper support the authors' conclusion very well.
**Reviewer #2 (Recommendations For The Authors):**
The paper by Ochesenbein describes the structural and functional analysis of *S. pombe* CAF-1 complex critical for DNA replication-coupled histone H3/H4 deposition. By using structural, biophysical, and biochemical analyses combined with genetic methods, the authors nicely showed that a large subunit of SpCAF1, SpPCF-1, consists of 5 structured domains with four connecting IDR domains. The ED domain with IDR nature binds to histone H3-H4 dimer with the conformational change of the other domain(s). SpCAF-1 binds to dsDNA by using the KER domain, but not the WHD domain. The experiments have been done with great care and a large amount of the data are highly reliable. Moreover, the results are clearly presented and convincingly written. The conclusion in the paper is very solid and will be useful for researchers who work in the field of chromosome biology.Major points:1. DNA binding of the KER mutant shown in Figures S3h and S3i, which was measured by the EMSA, looks similar to that of wild-type control in Figure S3f, which is different from the data in Figures 3b and 3e measured by the MST. The authors need a more precise description of the EMSA result of the KER mutant shown in Figures 3 and S3. The quantification of the EMSA result would resolve the point (should be provided).

A proposed by this reviewer, we performed quantification of all EMSA presented in Figure 3 and Figure S3. We quantified the signal of the free DNA band to calculate a percentage of bound DNA in each condition. All EMSA experiments were conducted in duplicate, allowing us to calculate an average value and standard deviation for each interaction. Representative curves and fitted values are reported below in the figure provided for the reviewer (panel a data for Pcf1_KER domain with two fitting models, panel b for the entire CAF-1 complexes and mutants, panel c for the isolated Pcf1_KER domains), all fitted values in panel d. Importantly, as illustrated in panel a, the complete model for a single interaction (complete KD model, dashed line curve) does not adequately fit the data. In contrast, a function incorporating cooperativity (Hill model) better accounts for the measured data (solid line curve). Consistently, we also used the Hill model to fit the binding curves measured with the MST technique. As also specified now in the text, the Hill model allows to determine an EC50 value (concentration of protein resulting in the disappearance of half of the free DNA band intensity) and a Hill coefficient value (representing cooperativity during the interaction) for each curve.

We measure a value of 3.4 ± 0.4 μM for the EC50 of SpCAF-1 WT, which is higher than the value measured by MST (0.7 ± 0.1 μM). Higher values were also calculated for all mutants and isolated Pcf1_KER domains compared to MST. These discrepancies could raise from the fact that the DNA concentration used in the two techniques were very different (20nM for MST experiments and 1μM for EMSA). Unlike the complete KD model, which includes in the calculation the DNA concentration (considered here as the "receptor"), the Hill model is fitted independently of this value. This model assumes that the “receptor” concentration is low compared to the KD. Here we calculate EC50 values on the same order of magnitude as the DNA concentration (low micromolar), The quantification obtained by EMSA is thus challenging to interpret. In contrast, values fitted by the MST measurements are more reliable since this limitation of low “receptor” concentration is correct.

Therefore, although measurements of EC50 and Hill coefficient from EMSA are reproducible, they may be confusing for quantifying apparent affinity values through EC50. Nevertheless, this quantitative analysis of EMSA, requested by the reviewer, has highlighted an interesting characteristic of the KER* mutant that is consistent across both methods: even though the EMSA pointed by the reviewer (Figures S3h and S3i compared to the wild-type control in Figure 3d and Figure S3f) show similar EC50 values, the binding cooperativity is different. Binding curves for the KER* mutants is no longer cooperative (Hill coefficient ~1), and this is observed for all KER* curves (isolated Pcf1_KER domain and the entire SpCAF-1 complex) with both methods, EMSA and MST. We thus decided to emphasize this characteristic of the KER* mutant in the text (page 9 line 30-32). “Importantly, this mutant also shows a lower binding cooperativity for DNA binding, as estimated by the Hill coefficient value close to 1, compared to values around 3 for the WT and other mutants.”

Since EMSA quantifications did not show a loss of “affinity” (as measured by the EC50 value) for the KER* mutants, compared to the WT contrary to MST measurements and because the DNA concentration was close to the measured EC50, we consider that EC50 values calculated by EMSA do not represent a KD value. If we add this quantification, we should discuss this point in detail. Thus, for sake of clarity, we prefer to put in the manuscript EMSA measurements as illustrations and qualitative validations of the interaction but not to include the quantification.

**Author response image 1. sa4fig1:** Quantitative analysis of interaction with DNA by EMSA. a: quantification of the amount of bound DNA for the Pcf1_KER domain (blue points with error bars). The fit with a KD model is shown as a dashed line, and the fit with a Hill model with a solid line. b: Examples of quantifications and fits (Hill model) for reconstituted SpCAF-1 WT and mutants. c: Examples of quantifications and fits (Hill model) for Pcf1_KER domains WT and mutant. d: EC50 values and Hill coefficients obtained for all EMSA experiments presented in Figure 3 and S3.

1. As with the cooperative DNA binding of CAF-1, it is very important to show the stoichiometry of CAF-1 to the DNA or the site size. Given a long alpha-helix of the KER domain with biased charges, it is also interesting to show a model of how the dsDNA binds to the long helix with a cooperative binding property (this is not essential but would be helpful if the authors discuss it).

We agree that having a molecular model for the binding of the KER helix to DNA would be especially interesting, but at this point, considering the accuracy of the tools currently at our disposal for predicting DNA-protein interactions, such a model would remain highly speculative.

1. Figure 5 shows nucleosome assembly by SpCAF-1. SpCAF-1-PIP* mutant produced a product with faster mobility than the control at 2 h incubation. How much amounts of SpCAF-1 was added in the reaction seems to be critical. At least a few different concentrations of proteins should be tested.

The slightly faster migration of the SpCAF-1-PIP* is not systematically reproduced and we observed in several experiments that the band corresponding to supercoiled DNA migrated slightly above or below the one for the complementation by the SpCAF-1-WT (see Author response image 2 below). Thus this indicates that after 2 hours incubation the supercoiling assay with the SpCAF-1-PIP* mutant compared to those achieved with the SpCAF-1-WT. To further document whether the WT or the PIP mutant are similar or not, we monitored difference of their nucleosome assembly efficiency by testing their ability to produce supercoiled DNA over shorter time, after 45 min incubation. Under these conditions, we reproducibly detected supercoiled forms at earlier times with SpCAF-1-WT when compared to the SpCAF-1-PIP* (see figure 5 and Author response image 2). These observations indicate that mutation in the PIP motif of Pcf1 affects the rate of supercoiling in a distinct manner when compared to the other mutations that dramatically impair SpCAF-1 capacity to promote supercoiling.

**Author response image 2. sa4fig2:** 

Minor points:1. Page 8, line 26 or Table 1 legend: Please explain what "EC50" is.

The definition of EC50, together with a reference paper for the Hill model have been added in the text page 8 lines 23-26, “The curves were fitted with a Hill model (Tso et al. 2018) with a EC50 value of 0.7± 0.1µM (effective concentration at which a 50% signal is observed) and a cooperativity (Hill coefficient, h) of 2.7 ± 0.2, in line with a cooperative DNA binging of SpCAF-1.”, in the Table 1 figure legend and in the method section (page 26).

1. Page 13, lines 9, 11: "*Xenopus*" should be italicized.

This is corrected

1. Page 14, second half: In *S. pombe*, the pcf1 deletion mutant is not lethal. It is helpful to mention the phenotype of the deletion mutant a bit more when the authors described the genetic analysis of various pcf1 mutants.

This point has been added on page 15, line 1.

1. Figure 1d and Figure S2a: Captions and labels on the X and Y axes are overlapped or misplaced.

This is corrected

1. Figure 5: Please add a schematic figure of the assay to explain how one can check the nucleosome assembly by looking at the form I, supercoiled DNAs.

A new panel has been added to Figure 5. This scheme depicts the supercoiling assay where supercoiled DNA (form I) is used as an indication of efficient nucleosome assembly. The figure legend has also been modified accordingly.

**Reviewer #3 (Public Review):**
Summary:The study conducted by Ouasti et al. is an elegant investigation of fission yeast CAF-1, employing a diverse array of technologies to dissect its functions and their interdependence. These functions play a critical role in specifying interactions vital for DNA replication, heterochromatin maintenance, and DNA damage repair, and their dynamics involve multiple interactions. The authors have extensively utilized various in vitro and in vivo tools to validate their model and emphasize the dynamic nature of this complex.Strengths:Their work is supported by robust experimental data from multiple techniques, including NMR and SAXS, which validate their molecular model. They conducted in vitro interactions using EMSA and isothermal microcalorimetry, in vitro histone deposition using *Xenopus* high-speed egg extract, and systematically generated and tested various genetic mutants for functionality in in vivo assays. They successfully delineated domain-specific functions using in vitro assays and could validate their roles to large extent using genetic mutants. One significant revelation from this study is the unfolded nature of the acidic domain, observed to fold when binding to histones. Additionally, the authors also elucidated the role of the long KER helix in mediating DNA binding and enhancing the association of CAF-1 with PCNA. The paper effectively addresses its primary objective and is strong.Weaknesses:A few relatively minor unresolved aspects persist, which, if clarified or experimentally addressed by the authors, could further bolster the study.1. The precise function of the WHD domain remains elusive. Its deletion does not result in DNA damage accumulation or defects in heterochromatin maintenance. This raises questions about the biological significance of this domain and whether it is dispensable. While in vitro assays revealed defects in chromatin assembly using this mutant (Figure 5), confirming these phenotypes through in vivo assays would provide additional assurance that the lack of function is not simply due to the in vitro system lacking PTMs or other regulatory factors.

Our work demonstrates that the WHD domain is important CAF-1 function during DNA replication. Indeed, the deletion of this domain lead to a synthetic lethality when combined with mutation of the HIRA complex, as observed for a null pcf1 mutant, indicating a severe loss of function in the absence of the WHD domain. We propose that these genetic interactions, previously reported in S. cerevisiae (Kaufman et al. MCB 1998; Krawitz et al. MCB 2002) are indicative of a defective histone deposition by CAF-1. Moreover, our work establishes that this domain is dispensable to prevent DNA damage accumulation and to maintain silencing at centromeric heterochromatin, indicating that the WHD domain specifies CAF-1 functions. Moreover, our work further demonstrates that, in contrast to the S. cerevisiae and human WHD domain, the *S. pombe* counterpart exhibits no DNA binding activity. We thus agree that the WHD domain may contribute to nucleosome assembly in vivo via PTMs or interactions with regulatory factors that may potentially lack in in vitro systems. However, addressing these aspects deserves further investigations beyond the scope of this article.

1. The observation of increased Pcf2-gfp foci in pcf1-ED* cells, particularly in mono-nucleated (G2phase) and bi-nucleated cells with septum marks (S-phase), might suggest the presence of replication stress. This could imply incomplete replication in specific regions, leading to the persistence of Caf1-ED*-PCNA factories throughout the cell cycle. To further confirm this, detecting accumulated single-stranded DNA (ssDNA) regions outside of S-phase using RPA as an ssDNA marker could be informative.

We cannot formally exclude that cells expressing the Pcf1-ED mutated form exhibit incomplete replication in specific regions, an aspect that would require careful investigations. However, the microscopy analysis (Fig. 6c and S6c) of this mutant showed no alteration in the cell morphology, including the absence of elongated cells compared to wild type, a hallmark of checkpoint activation caused by ssDNA (Enoch et al. Gene & Dev 1992). Therefore, investigating the consequences of the interplay between the binding of CAF-1 to PCNA and histones on the dynamic of DNA replication, is of particular interest but out of the scope of the current manuscript.

1. Moreover, considering the authors' strong assertion of histone binding defects in ED* through in vitro assays (Figure 2d and S2a), these claims could be further substantiated, especially considering that some degree of histone deposition might still persist in vivo in the ED* mutant (Figure 7d, viable though growth defective double ED*+hip1D mutants). For example, the approach, akin to the one employed in Fig. 6a (FLAG-IPs of various Pcf1-FLAG-tagged mutants), could also enable a comparison of the association of different mutants with histones and PCNA, providing a more thorough validation of their findings.

We have provided in the current manuscript data establishing how Pcf1 mutated forms interacted with PCNA (Fig. 6a, 6b). Regarding the interactions with histone H3-H4, the approach based on immunoprecipitation using various Pcf1-FLAG tagged mutants has been unsuccessful in our hands. Indeed, we were unable to obtain robust and reproducible interactions between Pcf1 or its various mutated form with H3-H4. This is likely because Co-IP approaches do not probe for direct interactions. Indirect interactions between Pcf1 and H3-H4 are potentially bridged by additional factors, including the two other subunits of CAF-1, Pcf2 and Pcf3, or Asf1. Therefore, we are not in a position to address in vivo the direct interactions between Pcf1 and histone H3-H4.

1. It would be valuable for the authors to speculate on the necessity of having disordered regions in CAF1. Specifically, exploring the overall distribution of these domains within disordered/unfolded structures could provide insightful perspectives. Additionally, it's intriguing to note that the significant disparities observed among mutants (ED*, PIP*, and KER*) in in vitro assays seem to become more generic in vivo, except for the indispensability of the WHD-domain. Could these disordered regions potentially play a crucial role in the phase separation of replication factories? Considering these questions could offer valuable insights into the underlying mechanisms at play.

We agree that the potential mechanistic role of partial disorder in CAF-1 is particularly interesting. Disordered regions of human CAF-1 have been reported to form nuclear bodies with liquid-liquid phase separation properties to maintain HIV latency (Ma et al EMBO J. 2021). As suggested, this raises the question of how disordered domains of Pcf1 could promote phase separation for replication factories, if such phenomenon happens in vivo. Moreover, numerous factors of the replisome also harbor disordered regions (Bedina, A. et al, 2013. Intrinsically Disordered Proteins in Replication Process. InTech. doi:10.5772/51673), adding complexity in disentangling experimentally such questions. We have added these elements at the end of the discussion in the revised manuscript (page 20, lines 23-29). “Such plasticity and cross-talks provided by structurally disordered domains might be key for the multivalent CAF-1 functions. Human CAF-1 has been reported to form nuclear bodies with liquid-liquid phase separation properties to maintain HIV latency (Ma et al. 2021). This raises the question of a potential role of the disordered domains of Pcf1, together with other replisome factor harbouring such disordered regions (Bedina 2013), in promoting phase separation of replication factories, if such phenomenon happens in vivo. Further studies will be needed to tackle these questions.”